# Causal Ordering for Structure Learning from Time Series

**Pedro P. Sanchez**\*  
*School of Engineering, University of Edinburgh, UK*  
*pedro.sanchez@ed.ac.uk*

**Damian Machlanski**\*  
*School of Engineering, University of Edinburgh, UK*  
*Causality in Healthcare AI Hub (CHAI), UK*  
*d.machlanski@ed.ac.uk*

**Steven McDonagh**  
*School of Engineering, University of Edinburgh, UK*  
*Causality in Healthcare AI Hub (CHAI), UK*  
*s.mcdonagh@ed.ac.uk*

**Sotirios A. Tsaftaris**  
*School of Engineering, University of Edinburgh, UK*  
*Causality in Healthcare AI Hub (CHAI), UK*  
*s.tsaftaris@ed.ac.uk*

**Reviewed on OpenReview:** *https://openreview.net/forum?id=hWuTzqggSd*

## Abstract

Predicting causal structure from time series data is crucial for understanding complex phenomena in physiology, brain connectivity, climate dynamics, and socio-economic behaviour. Causal discovery in time series is hindered by the combinatorial complexity of identifying true causal relationships, especially as the number of variables and time points grows. A common approach to simplify the task is the so-called ordering-based methods. Traditional ordering methods inherently limit the representational capacity of the resulting model. In this work, we fix this issue by leveraging multiple valid causal orderings, instead of a single one as standard practice. We propose DOTS (**D**iffusion **O**rdered **T**emporal **S**tructure), using diffusion-based causal discovery for temporal data. By integrating multiple orderings, DOTS effectively recovers the transitive closure of the underlying directed acyclic graph (DAG), mitigating spurious artifacts inherent in single-ordering approaches. We formalise the problem under standard assumptions such as stationarity and the additive noise model, and leverage score matching with diffusion processes to enable efficient Hessian estimation. Extensive experiments validate the approach. Empirical evaluations on synthetic and real-world datasets demonstrate that DOTS outperforms state-of-the-art baselines, offering a scalable and robust approach to temporal causal discovery. On synthetic benchmarks spanning $d=3-6$ variables, $T=200-5{,}000$ samples and up to three lags, DOTS improves mean window-graph $F1$ from 0.63 (best baseline) to 0.81. On the CausalTime real-world benchmark (*Medical*, *AQI*, *Traffic*; $d=20-36$), while baselines remain the best on individual datasets, DOTS attains the highest average summary-graph $F1$ while halving runtime relative to graph-optimisation methods. These results establish DOTS as a scalable and accurate solution for temporal causal discovery. Code is available at https://github.com/CHAI-UK/DOTS.

---

\*Equal contribution.

# 1 Introduction

Understanding cause-effect relationships from time series data is essential in fields like biology (Marbach et al., 2009), neuroscience (Friston et al., 2003), climate science (Runge et al., 2019), and economics (Pamfil et al., 2020), where uncovering how one event influences another can lead to valuable insights and better predictions. A key challenge in temporal causal discovery (Granger, 1969; Peters et al., 2013; Nauta et al., 2019; Runge, 2020) is the combinatorial complexity—there are many possible ways in which variables can influence each other, making it difficult to identify the true causal structure. The goal is to discover a temporal Directed Acyclic Graph (DAG), $\mathcal{G}$, representing these relationships. The core problem is illustrated in Figure 1.

Causal ordering approaches (Verma & Pearl, 1990; Friedman & Koller, 2003; Bühlmann et al., 2014; Rolland et al., 2022; Sanchez et al., 2023) offer a scalable alternative to direct graph estimation by constraining the search space, reducing it from a full adjacency matrix to a set of ordered permutations. While this reduction scales efficiently with respect to the number of variables and samples, it compromises representational power: a single causal ordering can imply extra edges, spurious artifacts, that are absent in the original DAG. In other words, committing to a single arbitrary ordering has an inherent downside: every node is deemed a potential ancestor of *all* subsequent nodes, creating spurious edges that must be pruned heuristically. To mitigate this, existing methods employ a two-step process, wherein a feature selection post-processing step prunes spurious artifacts introduced by the ordering.

Crucially, a DAG generally admits *multiple* valid orderings consistent with edge directions. Each ordering contributes complementary information about the underlying causal structure. By systematically generating and aggregating information from diverse orderings, we can filter out the spurious artifacts specific to any single, arbitrary ordering choice. As we aggregate more orderings, we converge towards the transitive closure ($\mathcal{G}^+$) of the underlying temporal DAG. $\mathcal{G}^+$ contains all true direct edges present in $\mathcal{G}$. While $\mathcal{G}^+$ also includes edges representing indirect causal pathways, it represents the stable, necessary ancestral relationships to recover the true graph. Therefore, harnessing several orderings increases both precision and recall and does so without incurring the combinatorial cost of full graph search. Our central insight is to embrace this multiplicity rather than fight it.

The benefit of multiple orderings hinges on access to a *diverse* collection of valid orderings. Naïve resampling or greedy heuristics tend to revisit near-duplicate permutations. We address this limitation by drawing on **denoising diffusion models** (Song & Ermon, 2019; Ho et al., 2020; Sanchez et al., 2023). Once trained to approximate the data score, a single diffusion network delivers Hessian estimates at *many noise scales*; applying a leaf-detection rule at each scale yields a fresh ordering. Because different scales emphasise different frequency bands of the data, the resulting orderings cover the search space more uniformly. Moreover, diffusion training amortises computation—after one network fit we can sample hundreds of orderings without

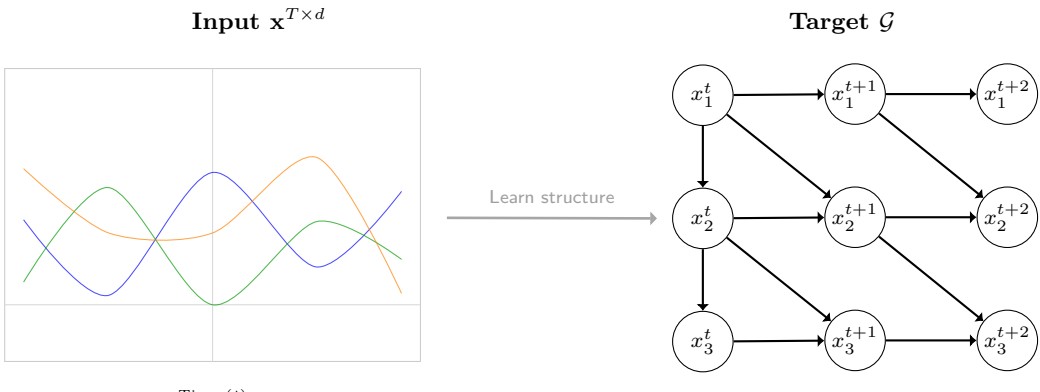

Figure 1: Temporal causal discovery estimates, from raw time series data (*left*), the underlying temporal causal DAG $\mathcal{G}$ (*right*).

retraining—making the approach viable for datasets that are large w.r.t. variables and samples. Finally, causal ordering approaches have not been extensively explored for temporal causal discovery, but their principles naturally extend to time-dependent data. Our focus on the temporal setting is motivated by two key advantages: (i) temporal constraints provide additional structural information that enhances ordering reliability through the temporal priority principle, and (ii) the temporal priority principle offers a natural filtering mechanism for spurious edges. The temporal dimension provides an inherent set of causal orderings, each contributing incremental insights.

Building on these ideas, we introduce DOTS (**D**iffusion-**O**rdered **T**emporal **S**tructure), a theoretically motivated ensemble-based approach for temporal causal discovery that systematically aggregates multiple diffusion-generated orderings. Our contributions are three-fold:

(i) We establish theoretically and empirically that aggregating multiple causal orderings to estimate the transitive closure ($\mathcal{G}^+$) provides a more robust foundation for temporal causal discovery than traditional single-ordering approaches, effectively filtering spurious edges arising from arbitrary ordering choices.

(ii) We introduce DOTS, a novel algorithm leveraging multi-scale diffusion models to efficiently generate the diverse set of causal orderings required for robust aggregation in the temporal domain.

(iii) We provide extensive benchmarks, adapting several static ordering methods for temporal data and demonstrating the superior performance of the multi-ordering strategy implemented by DOTS against both these adapted methods and state-of-the-art temporal causal discovery baselines on synthetic and real-world datasets.

The rest of the paper is organised as follows. Section 2 reviews notation and identifiability assumptions. Section 3 establishes the theoretical foundations of multi-ordering aggregation, and Section 4 presents the DOTS algorithm in detail. Section 5 shows experimental results. Relevant literature is discussed in Section 6. Section 7 concludes the paper.

## 2 Preliminaries

### 2.1 Notation

We present in Table 1 a description of all the symbols in the manuscript.

### 2.2 Problem Definition

We aim to discover the causal structure among $d$ variables arranged in a temporal setting. Let $\mathbf{x}^t \in \mathbb{R}^d$ denote a vector-valued random variable at time $t$, with components $\left(\mathrm{x}_1^t, \ldots, \mathrm{x}_d^t\right)$. A *temporal directed acyclic graph* (temporal DAG) $\mathcal{G}^t$ then describes the causal relationships among these variables, where each node corresponds to a component $\mathrm{x}_i^t$, and the directed edges represent causal links between variables at the same time or across time steps (Eichler, 2012). We now introduce standard assumptions for causal discovery from time series:

*Assumption* 1 (Temporal DAG). The true causal structure of the data can be represented by a temporal DAG $\mathcal{G}^t$. This graph includes lagged edges of the form $\mathrm{x}_i^{t-\tau} \to \mathrm{x}_j^t$ (for $\tau > 0$) and contemporaneous edges $\mathrm{x}_i^t \to \mathrm{x}_j^t$. This assumption effectively captures the *temporal priority principle* (Hume, 1904; Rankin & McCormack, 2013; Assaad et al., 2022) where causes precede effects in time. For any pair $\left(\mathrm{x}_i^{t-\tau}, \mathrm{x}_j^t\right)$, an edge $\mathrm{x}_i^{t-\tau} \to \mathrm{x}_j^t$ indicates that the value of $\mathrm{x}_i$ at time $(t-\tau)$ influences $\mathrm{x}_j$ at time $t$. This forbids backward-in-time causation and helps simplify structure learning since the search space is smaller.

*Assumption* 2 (Stationarity). The causal relationships in $\mathcal{G}^t$ remain invariant over time; that is, both the causal links and their strengths remain unchanged for all values of $t$.

*Assumption* 3 (Time Series Models with Independent Noise (TiMINo)). The structural causal model follows a temporal additive noise formulation:

$$\mathrm{x}_j^t = f_j\left(Pa_{\mathcal{G}^t}(\mathrm{x}_j^t)\right) + \epsilon_j^t, \tag{1}$$

Table 1: Summary of notation used throughout the paper.

| Symbol | Meaning / role in the paper |
| --- | --- |
| $\mathbf{x} \in \mathbb{R}^d$ | Random vector of $d$ variables; component $i$ is $\mathrm{x}_i$. |
| $x_i^t, \mathbf{x}^t$ | Value / vector at time index $t$. |
| $d$ | Number of variables (dimensionality). |
| $T$ | Number of observed time steps. |
| $\tau$ | Time lag; $\mathrm{x}_i^{t-\tau} \to \mathrm{x}_j^t$. |
| $\tau_{\max}$ | Maximum lag included in $\mathbf{A}$. |
| $k \in \{0, \ldots, k_{\max}\}$ | Diffusion (noise-scale) timestep. |
| $k_{\max}$ | Final diffusion step (fully noised). |
| $\pi$ | Causal ordering (topological permutation). |
| $\pi_i$ | Variable at position $i$ in ordering $\pi$. |
| $\mathcal{G}$ | (Temporal) causal DAG. |
| $\mathcal{G}^+$ | Transitive closure of $\mathcal{G}$. |
| $\mathbf{A} \in \{0,1\}^{d(\tau_{\max}+1) \times d(\tau_{\max}+1)}$ | Temporal adjacency matrix to be learned. |
| $p(\mathbf{x})$ | Data distribution (density). |
| $\boldsymbol{\epsilon}_\theta(\mathbf{x}, k)$ | Neural network estimating $\nabla_{\mathbf{x}} \log p(\mathbf{x})$ at scale $k$. |
| $\tilde{\mathbf{x}}^k$ | Noisy version of $\mathbf{x}$ at diffusion step $k$. |
| $q(\tilde{\mathbf{x}}^k \mid \mathbf{x}, k)$ | Forward noising distribution. |
| $f_j$ | Structural function generating $\mathrm{x}_j$ from its parents. |
| $\epsilon_j^t$ | Independent noise term for $\mathrm{x}_j^t$. |
| $Pa_{\mathcal{G}^t}(\mathrm{x}_j^t)$ | Parent set of $\mathrm{x}_j^t$ in the temporal DAG. |
| $Ch(\mathrm{x}_j)$ | Children of node $\mathrm{x}_j$. |
| $\boldsymbol{H}_{i,j}\big(\log p(\mathbf{x})\big)$ | $(i,j)$-entry of Hessian of $\log p(\mathbf{x})$ (score Jacobian). |
| $E(\pi)$ | Edge set implied by ordering $\pi$. |
| $E_{\mathrm{agg}}^{(m)}$ | Intersection of edge sets from $m$ sampled orderings. |
| $\mathcal{T}$ | Set of temporally valid edges (causes precede effects). |
| $E_{\mathrm{agg}}^{(m,T)}$ | Aggregated edge set restricted to $\mathcal{T}$. |
| $W_{ij}$ | Fraction of sampled orderings where edge $i \to j$ appears (vote matrix). |
| $\tilde{A}_{ij}$ | Soft transitive-closure entry after thresholding $W$. |
| $\theta$ | Threshold for vote-matrix aggregation ($0 < \theta \le 1$). |
| $\alpha_k$ | Variance-preserving coefficient in the forward diffusion process. |

where $Pa_{\mathcal{G}^t}(\mathrm{x}_j^t)$ denotes the set of parent variables of $\mathrm{x}_j^t$ in $\mathcal{G}^t$, $f_j$ is a nonlinear function, and $\epsilon_j^t$ is an independent noise term. TiMINo (Peters et al., 2013) essentially extends the additive noise model (ANM) framework to time series.

*Assumption* 4 (Causal Sufficiency). All common causes of observed variables are measured; that is, there are no unobserved confounders that influence multiple components simultaneously.

*Assumption* 5 (Noise Distribution Regularity). The exogenous noise terms $\epsilon_j^t$ in Equation 1 have densities $p^{\mathrm{u}}$ such that $\frac{\partial^2 \log p^{\mathrm{u}}}{\partial x^2}$ is constant. This includes Gaussian noise as a special case but also encompasses other distributions with quadratic log-densities. This regularity condition is required for the score-matching approach to satisfy the leaf-detection criterion in Equation 3 and extends the assumptions of Sanchez et al. (2023) to the temporal domain using the TiMINo identifiability framework (Peters et al., 2013).

## 2.3 Objective

Given an observational multivariate time series dataset $\mathbb{D} \in \mathbb{R}^{T \times d}$ containing $T$ timesteps of $d$ variables, our goal is to learn the temporal adjacency structure $\mathbf{A} \in \{0,1\}^{d(\tau_{\max}+1) \times d(\tau_{\max}+1)}$, where $\tau_{\max}$ denotes the maximum lag that is being captured in the adjacency matrix. Each entry of $\mathbf{A}$ encodes whether there is a directed edge from $\mathrm{x}_i^{t-\tau}$ to $\mathrm{x}_j^t$ for $0 \le \tau \le \tau_{\max}$. Note that $\tau = 0$ indicates a contemporaneous link $\mathrm{x}_i^t \to \mathrm{x}_j^t$.

## 2.4 Key Identifiability Results

Under stationarity, the additive noise assumption, and causal sufficiency, Peters et al. (2013) show that temporal causal relationships become identifiable if the data follow a restricted structural equation model in which each noise term is statistically independent and no directed cycles exist within a single time slice. Specifically, their *Time Series Models with Independent Noise* (TiMINo) framework demonstrates that both lagged and instantaneous effects can be recovered uniquely, provided the functional form and noise distributions meet certain identifiability criteria (e.g. linear non-Gaussian or nonlinear Gaussian).

## 2.5 Causal ordering

Causal search over the space of DAGs is an NP-hard problem (Chickering, 1996). Traditional approaches leverage heuristic search strategies to navigate the combinatorial space of potential DAG structures. Order-based search offers a simpler and more effective alternative. By shifting the search from graph structures to node orderings, the strategy exploits the fact that, for a given ordering, identifying the highest-scoring network is not NP-hard. Such causal ordering approaches reduce the search space and inherently satisfy the acyclicity constraints. This bypasses the need for explicit acyclicity checks during the search. These methods find a particular causal ordering of the nodes, i.e. a list of nodes such that a node in the ordering can be a parent only of the nodes appearing after it in the exact ordering. Causal ordering is also known as topological ordering or a causal list in the causal discovery literature (Peters et al., 2017). Formally, causal ordering of a DAG $\mathcal{G}$ is defined as a non-unique permutation $\pi$ of $d$ nodes. Hence, a given node in $\pi$ always appears before its descendants in the list. Or more formally, $\pi_i < \pi_j \iff j \in De(x_i)$ where $De(x_i)$ are the descendants of the *ith* node in $\mathcal{G}$ (Appendix B in Peters et al. (2017)).

## 2.6 Causal ordering via score matching

Rolland et al. (2022) propose that the score of an ANM with distribution $p(\mathbf{x})$ can be used to estimate the causal ordering by finding leaves. Leaves are nodes of DAG $\mathcal{G}$ that do not have children. Rolland et al. (2022) propose a method to find leaves based on the derivative of the ANM log density (also called *score*). An analytical expression for the score of an ANM from Equation 1 is

$$\nabla_{x_j} \log p(\mathbf{x}) = \frac{\partial \log p^{\mathrm{u}} \left(x_j - f_j\right)}{\partial x_j} - \sum_{i \in Ch(x_j)} \frac{\partial f_i}{\partial x_j} \frac{\partial \log p^{\mathrm{u}} \left(x_i - f_i\right)}{\partial x}, \tag{2}$$

where $Ch(x_j)$ denotes the children of $x_j$. Using this analytical equation of $\nabla_{x_j} \log p(\mathbf{x})$, Rolland et al. (2022) derive the following condition used to find leaf nodes. Given a nonlinear ANM with a noise distribution $p^{\mathrm{u}}$ and a leaf node $j$, assuming that $\frac{\partial^2 \log p^{\mathrm{u}}}{\partial x^2} = a$, where $a$ is a constant, then

$$\mathrm{Var}_{\mathcal{D}} \left[ \boldsymbol{H}_{j,j}(\log p(\mathbf{x})) \right] = 0. \tag{3}$$

This rule is based on the score's Jacobian (or Hessian of the log distribution). $\boldsymbol{H}_{j,j}(\log p(\mathbf{x}))$ is used in Rolland et al. (2022) to propose a causal ordering algorithm that iteratively finds and removes leaf nodes from the dataset. Rolland et al. (2022) re-compute the score's Jacobian with a kernel-based estimation method at each iteration.

## 2.7 Approximating the score's Jacobian via diffusion training

Instead of computing $\log p(\mathbf{x})$ via a kernel-based estimation (Li & Turner, 2018; Rolland et al., 2022), we follow Sanchez et al. (2023) and estimate the score's Jacobian with diffusion models (Song & Ermon, 2019; Ho et al., 2020). This estimation is based on a diffusion process that progressively corrupts $\mathbf{x}$ with Gaussian noise over timesteps $k \in \{0, \dots, k_{\max}\}$. Let $\tilde{\mathbf{x}}^k$ be the noisy version of $\mathbf{x}$ at diffusion step $k$. A neural network $\boldsymbol{\epsilon}_\theta(\tilde{\mathbf{x}}^k, k)$ is trained to denoise $\tilde{\mathbf{x}}^k$ back to $\mathbf{x}$, thereby approximating the true score $\nabla_{\mathbf{x}} \log p(\mathbf{x})$. Formally, this can be written as:

$$\mathbb{E}_{\mathbf{x} \sim p(\mathbf{x}), \ \tilde{\mathbf{x}}^k \sim q(\tilde{\mathbf{x}}^k | \mathbf{x}, k)} \left\| \boldsymbol{\epsilon}_\theta\left(\tilde{\mathbf{x}}^k, k\right) - \nabla_{\tilde{\mathbf{x}}^k} \log p\left(\tilde{\mathbf{x}}^k \mid k\right) \right\|^2,$$

where $q(\tilde{\mathbf{x}}^k \mid \mathbf{x}, k)$ defines the forward noising process. Once trained, $\boldsymbol{\epsilon}_\theta$ effectively yields $\nabla_{\mathbf{x}} \log p(\mathbf{x})$ at various noise scales $k$, which can be used to estimate the Hessian for causal discovery. Noise at multiple scales explores regions of low data density (Song & Ermon, 2019).

The score's Jacobian can be approximated by learning the score $\boldsymbol{\epsilon}_\theta$ with denoising diffusion training of neural networks and back-propagating (Rumelhart et al., 1986)[1] from the output to the input variables. The quantity can be written, for an input data point $\boldsymbol{x} \in \mathbb{R}^d$, as

$$\boldsymbol{H}_{i,j} \log p(\boldsymbol{x}) \approx \nabla_{i,j} \boldsymbol{\epsilon}_\theta(\boldsymbol{x}, k), \tag{4}$$

where $\nabla_{i,j} \boldsymbol{\epsilon}_\theta(\boldsymbol{x}, k)$ means the *ith* output of $\boldsymbol{\epsilon}_\theta$ is backpropagated to the *jth* input. The diagonal of the Hessian in Equation 4 can be used to find leaf nodes as in Equation 3. We use masking (Sanchez et al., 2023) to iteratively find and remove leaf nodes, *without* retraining the score.

## 3   Theoretical motivation: temporal structure from multiple causal orderings

Causal ordering methods, illustrated in Figure 2, traditionally rely on a single ordering to infer the full DAG in causal discovery. A single causal graph does not contain sufficient information to reliably infer the DAG, since each node is considered a cause for all subsequent nodes. Therefore, pruning methods are used to remove spurious edges (Bühlmann et al., 2014). With infinite data a perfect conditional-independence oracle could delete the spurious edges and retain the true ones. In practice we face finite samples, noisy tests and high dimensionality. Starting from an over-dense candidate set inflates both kinds of statistical error: *(i) false positives* remain whenever a test *fails* to reject a truly absent edge (type-II error), and *(ii) false negatives* appear when a test mistakenly deletes a true edge (type-I error) because that edge co-varies with many irrelevant ancestors in the initial ordering. Hence a single ordering often yields a fragile estimate whose quality varies wildly with sample size and noise level. In contrast, leveraging multiple causal orderings provides a richer representation of the underlying structure. Exploiting multiple valid causal orderings from data naturally follows from the fact that a given DAG typically admits more than one linear ordering, consistent with its structure.

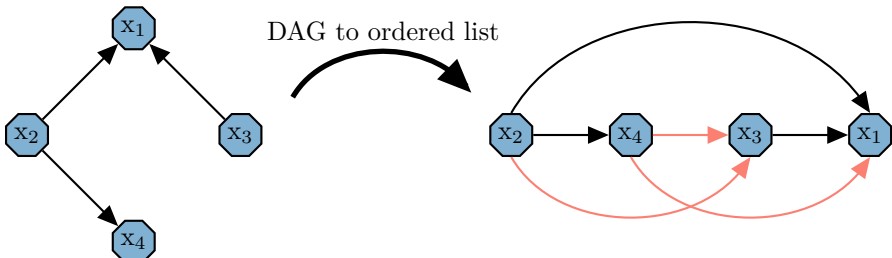

Figure 2: A DAG can be represented as an ordered list following the causal direction. A node in the ordering can cause any subsequent node. Searching over the space of permutations is more efficient than searching over the 2D space of matrices. However, topologically sorting nodes reduces the amount of information in the representation of causal relationships.

Rather than committing to a single topological sort, estimating multiple valid orderings can offer a more complete representation of ancestor–descendant relationships. In doing so, we can aggregate local adjacency constraints from each ordering and thereby recover, asymptotically, the *transitive closure* of the DAG—the minimal set of edges that preserves causal reachability. This perspective avoids over-specifying the order of variables that are not causally linked and reduces the risk of introducing extra edges that do not exist in the underlying temporal DAG.

---

[1]The Jacobian of a neural network can be efficiently computed with auto-differentiation libraries such as functorch (Horace He, 2021).

### 3.1 Recovering DAG structure from a complete set of causal orderings

We now investigate how the topological orderings of a DAG relate to its underlying structure. The central takeaway is that the collection of all topological orderings of a DAG uniquely determines the *transitive closure* $\mathcal{G}^+$, which is a robust representation of the graph's reachability structure. Topological orderings capture the reachability relation of the graph—a partial order—but multiple DAGs can share the same reachability relation and, consequently, the same set of topological orderings.

In a DAG $\mathcal{G}$, every topological ordering is a linear extension of the reachability relation $R$. A key insight from order theory is formalised in *Hiraguchi's theorem* (Hiraguchi, 1955), which establishes that any finite partial order can be exactly recovered as the intersection of all its linear extensions (i.e. total orders consistent with it). In our setting, this implies that aggregating multiple topological orderings of a DAG asymptotically recovers its transitive closure. This idea is formalized in the following proposition:

**Proposition 1** (Reconstruction of the Transitive Closure). *Let $G = (V, E)$ be a DAG, and let $\mathcal{L}$ be the set of all its topological orderings. Define a binary relation $\prec$ on $V$ by:*

$$x \prec y \iff x \text{ appears before } y \text{ in every } \pi \in \mathcal{L}.$$

*Then:*

*(i) $\prec$ is a strict partial order on $V$ (irreflexive and transitive).*

*(ii) For all $x, y \in V$,*

$$x \prec y \iff \text{ there exists a directed path from } x \text{ to } y \text{ in } G \text{ with } x \neq y.$$

*Thus, $\prec$ matches the edges of the transitive closure $\mathcal{G}^+$.*

*(iii) Consequently, aggregating all topological orderings in $\mathcal{L}$ recovers $\mathcal{G}^+$, but not necessarily the original DAG $G$.*

*Justification.* We show part (2) by establishing the equivalence:

- ($\Rightarrow$): If there is a directed path from $x$ to $y$ in $G$ with $x \neq y$, then every topological ordering $\pi \in \mathcal{L}$ must place $x$ before $y$ to respect the direction of edges. Hence, $x \prec y$.

- ($\Leftarrow$): Suppose $x \prec y$ but no directed path from $x$ to $y$ exists in $G$ with $x \neq y$. Since $G$ is a DAG and no path exists from $x$ to $y$, adding the edge $(y, x)$ to $G$ does not introduce a cycle (otherwise, a path from $x$ to $y$ would exist, contradicting the assumption). In this modified DAG, there exists a topological ordering with $y$ before $x$, contradicting $x \prec y$. Thus, a directed path from $x$ to $y$ must exist in $G$.

This shows that $\prec$ corresponds to the strict reachability relation, i.e. the edges of $\mathcal{G}^+$. Since distinct DAGs can share the same $\mathcal{G}^+$, the original $G$ cannot be uniquely recovered from $\mathcal{L}$. $\qquad\square$

**Example 1.** *Let $G_1 = (V, E_1)$ with $V = \{a, b, c\}$ and $E_1 = \{(a, b), (b, c)\}$, and let $G_2 = (V, E_2)$ with $E_2 = \{(a, b), (b, c), (a, c)\}$. Both DAGs share the same set of topological orderings: $\{a, b, c\}$. In $G_1$, the ordering respects $a \to b$ and $b \to c$; in $G_2$, the additional edge $(a, c)$ is consistent with the order. The transitive closure for both is $G_1^+ = G_2^+ = (V, \{(a, b), (b, c), (a, c)\})$. Thus, from the common ordering alone, we recover $G_1^+$ (or $G_2^+$) but cannot distinguish between $G_1$ and $G_2$.*

Aggregating all topological orderings of a DAG $G$ yields its transitive closure $\mathcal{G}^+$ because the relation $\prec$ captures all pairs of nodes connected by a directed path. However, since distinct DAGs can share the same transitive closure, the original edge set $E$ remains ambiguous. This highlights a fundamental limit: while topological orderings reveal the reachability structure of the graph, they do not specify the precise topology of the DAG. It is crucial to understand that the transitive closure $\mathcal{G}^+$ includes both direct edges from the original DAG $\mathcal{G}$ and indirect relationships (edges representing multi-step causal pathways). While this

means that $\mathcal{G}^+$ is not identical to the true causal DAG $\mathcal{G}$, it serves as a robust intermediate representation that captures all ancestral relationships. The subsequent pruning step (Section 4.4) is specifically designed to distinguish between direct and indirect relationships, refining $\mathcal{G}^+$ to recover the sparse structure of $\mathcal{G}$. Nevertheless, recovering the transitive closure $\mathcal{G}^+$ is a stronger claim than what has been previously achieved by single ordering methods (CAM, SCORE, DAS, NoGAM, DiffAN). A single ordering recovers only a weak approximation of the causal structure because, for any given position in the ordering, all subsequent positions are considered as potential descendants, introducing spurious edges in addition to possible indirect edges.

## 3.2 Ordering aggregation and recovery of temporal structure

In causal discovery, we do not know a priori the total number of valid causal orderings that a resulting DAG will admit. The number of causal orderings is not only related to the number of variables but also to the edge topology and density. Therefore, we next explore how aggregating a subset of the total orderings improves structure estimation. Empirical evidence, shown in Figure 3, suggests that enumerating all topological sorts is unnecessary to achieve strong performance, as a relatively small subset of randomly sampled orderings can suffice for accurate structure recovery.

To formalize the benefit of multiple orderings, assume that for each valid causal ordering $\pi$ of a DAG $\mathcal{G}$, we obtain an edge set $E(\pi)$ that corresponds to the directed edges implied by that ordering. Define the aggregated edge set over the intersection of $m$ orderings as

$$E_{\text{agg}}^{(m)} = \bigcap_{i=1}^{m} E(\pi_i).$$

Then, under the assumption that each valid ordering provides complementary information about the true ancestral relations, following Proposition 1, we have

$$\lim_{m \to M} E_{\text{agg}}^{(m)} = \mathcal{G}^+,$$

where $\mathcal{G}^+$ denotes the transitive closure of the true edge set $E$ of $\mathcal{G}$ and where $M$ is the number of possible orderings for $\mathcal{G}$.

To validate this idea, we generate random temporal DAGs and use Kahn's algorithm (Kahn, 1962) to list all possible orderings with topological sorting. Then, we estimate the transitive closure for each DAG and compare it with the estimated $E_{\text{agg}}^{(m)}$ as we increase $m$ for each DAG. We compare the estimated and true transitive closure via F1 score. In Figure 3, we show the percentage of orderings $\frac{m}{M} * 100$ used with respect to the F1 score. For our data, we observe that $\sim 40\%$ of all orderings is generally enough to recover the transitive closure with high F1 score.

## 3.3 Incorporating Temporal Constraints

Temporal constraints are critical when recovering causal structure from time series data. Following the *temporal priority principle*; causes precede effects, in time. This principle forbids backward-in-time causation and simplifies structure learning.

To incorporate this constraint into edge aggregation, we restrict the aggregated edge set to include only temporal edges satisfying $t - \tau < t$. Formally, let

$$\mathcal{T} = \left\{ (x_i^{t-\tau}, x_j^t) \mid \tau \geq 0, \ t - \tau < t \right\}$$

denote the set of all temporally valid edges. Then, define the temporally constrained aggregated edge set as

$$E_{\text{agg}}^{(m,T)} = E_{\text{agg}}^{(m)} \cap \mathcal{T}.$$

This filtering ensures that only edges adhering to the temporal priority principle are retained, thereby excluding any spurious backward-in-time connections and further enhancing the reliability of the recovered DAG. Empirically, as illustrated in Figure 3, adding the temporal constraint decreases the number of causal orderings required to estimate the causal structure for a given F1 score.

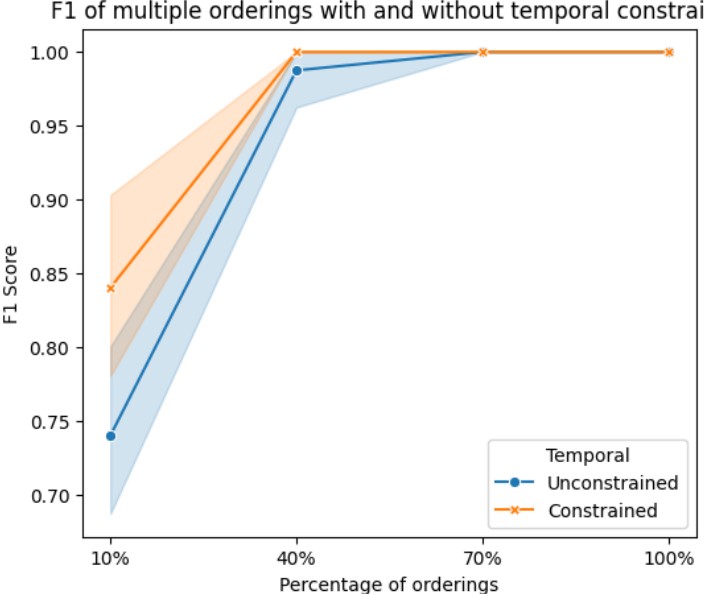

Figure 3: Impact of multiple causal orderings on DAG recovery. A single (or few) ordering (left) may include extra or spurious edges, whereas aggregating multiple orderings (right) more accurately recovers the full transitive closure of the underlying DAG.

In summary, integrating temporal constraints into the aggregation of multiple causal orderings is both a natural extension of, and an efficient strategy for, structure learning in time series data. By leveraging the inherent temporal priority principle—where causes always precede effects—we effectively filter and refine the aggregated edge set, ensuring that only temporally valid connections are retained. This dual approach not only enhances the robustness of the recovered causal structure by capturing complementary ancestral information across orderings but also streamlines the learning process, as it reduces the effective search space and mitigates spurious dependencies.

While our empirical validation demonstrates the theoretical potential of aggregating multiple causal orderings for recovering the transitive closure of a DAG, it is important to note that this evaluation was performed on known graphs where all valid causal orderings were available and the total number of orderings was predetermined. Orderings were selected uniformly at random from this complete set. We further note that for practical applications, an algorithm may generate very similar orderings, thereby limiting the diversity necessary for robust structure recovery. In the following section, we explore methods to induce sufficient variability in the generated orderings, enabling the aggregation process to remain both effective and efficient in recovering the true causal structure.

## 4  A Diffusion-Based Approach for Temporal Discovery

We introduce DOTS, a method that utilises diffusion processes to recover multiple valid causal orderings in temporal data. Our approach, illustrated in Figure 4, integrates both a frequency domain perspective and multi-scale causal ordering to capture the complex structure of temporal relationships. We refer to our notation (Section 2.1), distinguishing between time lags ($\tau$), diffusion timesteps ($k$), and the indices for causal orderings ($\pi$).

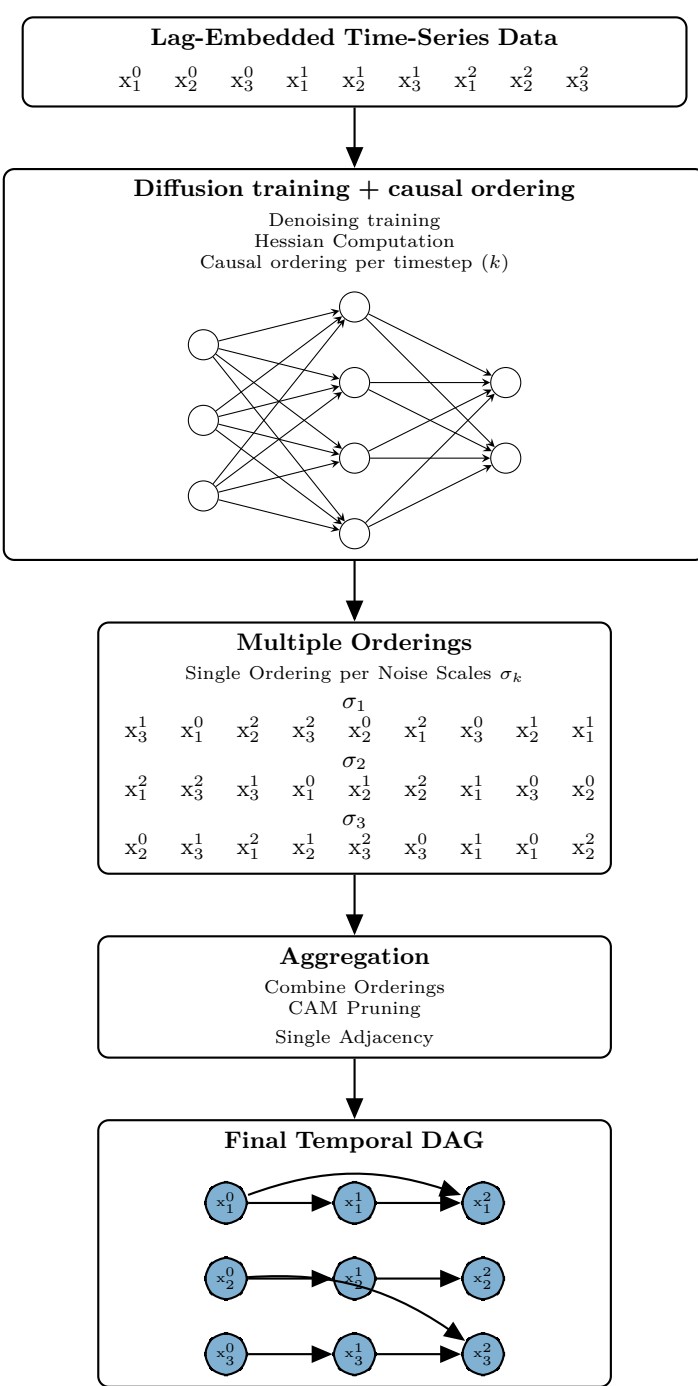

Figure 4: DOTS pipeline for temporal causal discovery. We start with lag-embedded time-series data, apply diffusion-based single-order discovery, then extend to multiple orderings and aggregate them. The final temporal DAG below shows an example with three variables over three timesteps.

## 4.1 Why do diffusion steps capture different frequency components?

Different diffusion steps $k$ capture distinct aspects of the data, which can be understood from a frequency perspective. Consider a forward diffusion process (e.g. DDPM (Ho et al., 2020)), which decomposes each observation at step $k$ as

$$x_k = \sqrt{\alpha_k}\, x_0 + \sqrt{1 - \alpha_k}\, \epsilon, \quad \epsilon \sim \mathcal{N}(0, \boldsymbol{I}).$$

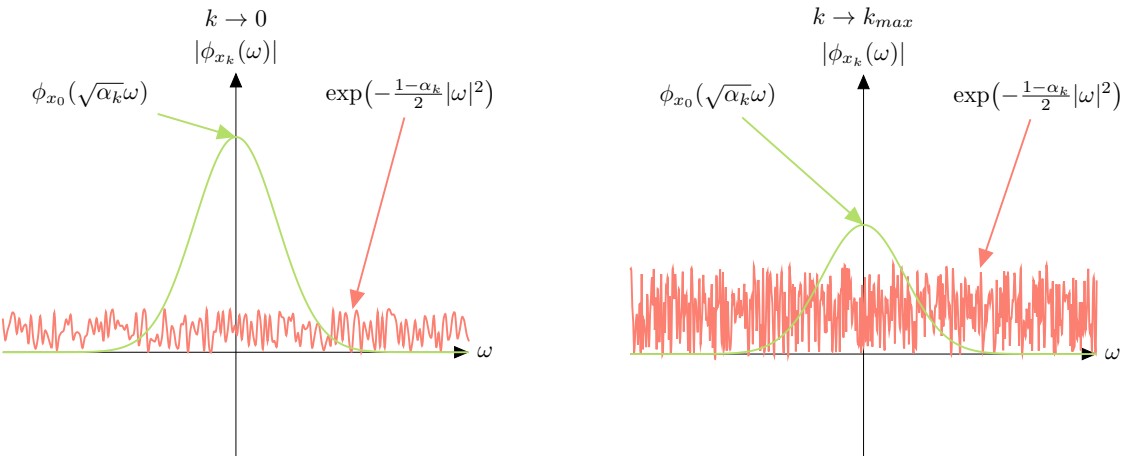

Figure 5: **Frequency emphasis of diffusion steps.** A forward diffusion step decomposes $x_k = \sqrt{\alpha_k}x_0 + \sqrt{1 - \alpha_k}\,\epsilon$. As can be seen in the Fourier domain, high values of $k$ emphasize learning of low-frequency while low values of $k$ force the network to focus on high-frequency components.

Since $x_0$ and $\epsilon$ are independent, the characteristic function of $x_k$ becomes:

$$\phi_{x_k}(\omega) = \phi_{\sqrt{\alpha_k}x_0}(\omega) \cdot \phi_{\sqrt{1-\alpha_k}\epsilon}(\omega) = \phi_{x_0}(\sqrt{\alpha_k}\omega) \cdot \phi_\epsilon(\sqrt{1 - \alpha_k}\omega).$$

For Gaussian $\epsilon$, we have $\phi_\epsilon(\omega) = \exp(-\frac{1}{2}|\omega|^2)$, so the noise component contributes $\exp(-\frac{1-\alpha_k}{2}|\omega|^2)$, which acts as a low-pass filter with bandwidth inversely related to $\sqrt{1 - \alpha_k}$. As $k$ increases, $\alpha_k$ decreases, making the filter narrower and emphasizing lower frequencies. This creates a natural multi-scale decomposition where large $k$ values capture coarse, low-frequency structure while small $k$ values preserve fine, high-frequency details.

Figure 5 schematically illustrates how each $k$ emphasizes different frequency components; large $k$ reveals coarse causal links and small $k$ highlights finer edges. This multi-scale view enables the network to focus on spectral components most impacted by noise at each scale. Similar conclusions were drawn based on observations in the imaging domain (Kascenas et al., 2023).

### 4.2 Lag-Embedded Representation of Time Series

The diffusion model is trained on the lag-embedded representation of the time series. Lag-embedding is a common technique in dynamical systems theory to capture temporal dependencies (Takens, 1981). Let $\boldsymbol{X} = [\mathbf{x}^1, \ldots, \mathbf{x}^T]^\top \in \mathbb{R}^{T \times d}$ denote the raw sequence with $d$ variables and $T$ time steps. We construct a *lag-embedded* matrix

$$\mathbb{D} = \left[\mathbf{x}^t,\ \mathbf{x}^{t-1}, \ldots, \mathbf{x}^{t-\tau_{\max}}\right]_{t=\tau_{\max}}^{T-1} \in \mathbb{R}^{(T-\tau_{\max}) \times d(\tau_{\max}+1)}.$$

Each column of $\mathbb{D}$ now refers to a specific variable–lag pair $x_i^{t-\tau}$. This representation feeds the diffusion network; temporal precedence is enforced later when we discard any edge that points backwards in time.

**Incorporating temporal information** For temporal data with lags $\tau$, each variable is indexed as $x_i^t$, and potential edges include both lagged ($x_i^{t-\tau} \to x_j^t$) and contemporaneous ($x_i^t \to x_j^t$) relationships. A single diffusion model is trained on the lag-embedded dataset $\mathbb{D} \in \mathbb{R}^{(T-\tau_{max}) \times (d \times \tau_{max})}$, and the leaf-finding process is applied in the same manner, ensuring that each node is treated as a time-indexed variable. This strategy enforces a temporal DAG that captures both lagged and instantaneous dependencies.

### 4.3 Multi-Scale Causal Orderings

Each diffusion step $k$ corresponds to a distinct noise regime, and consequently, the Hessian $\boldsymbol{H}_{\mathbf{x}} \log p(\mathbf{x})$ computed at each $k$ reveals different adjacency constraints. Large $k$ values tend to highlight broad, low-frequency cause–effect relationships, while small $k$ values accentuate fine-grained, high-frequency interactions.

**Multiple orderings at different $k$.** Instead of relying on a single noise scale, we execute a causal ordering algorithm at several discrete diffusion steps $\{k_1, \ldots, k_S\}$. Each execution yields a *causal ordering*, denoted by $\pi$, which reflects the partial order implied by that particular $k$. This process generates multiple orderings $\pi_1, \ldots, \pi_S$, thereby capturing the multi-scale structure inherent in the data. We then identify leaf nodes via the diagonal of the Hessian, following the approach of Rolland et al. (2022) and (Sanchez et al., 2023).

**Causal ordering with a Hessian from diffusion training.** After training a diffusion model $\boldsymbol{\epsilon}_\theta(\mathbf{x}, k)$, we approximate the partial derivatives as

$$\boldsymbol{H}_{i,j} \log p(\mathbf{x}) \;\approx\; \frac{\partial}{\partial x_j}\Big[\boldsymbol{\epsilon}_\theta(\mathbf{x}, k)\Big]_i,$$

for each $k$. The diagonal entries $\boldsymbol{H}_{i,i}$ exhibit lower variance for leaf nodes than for non-leaf nodes (Rolland et al., 2022; Sanchez et al., 2023). To identify a leaf node, we: (i) Estimate $\boldsymbol{H}(\mathbf{x}, k)$ on a mini-batch of data $\mathbb{D}$. (ii) Identify the variable $x_\ell$ with the lowest diagonal variance $\mathrm{Var}\big[\boldsymbol{H}_{\ell,\ell}\big]$. (iii) Remove $x_\ell$ from the distribution by masking out the variables in the input as done with DiffAN masking (Sanchez et al., 2023). This procedure is repeated until all variables are assigned an order, yielding a complete causal ordering $\pi$. Repeating this process for each chosen $k$ produces the set $\{\pi_1, \ldots, \pi_S\}$.

After this procedure, the set $\{\pi_1, \ldots, \pi_S\}$ can be aggregated as described in Section 4.4. In essence, each $\pi_s$ represents a valid causal ordering that reflects the partial order constraints emphasized at its respective diffusion timestep $k_s$. By uniting these multi-scale perspectives, the DOTS algorithm produces a final temporal DAG that captures both coarse (low-frequency) and fine (high-frequency) causal interactions.

---

**Algorithm 1** Estimating Multi-Scale Causal Orderings.

---

**Require:**
  $\mathbb{D} \in \mathbb{R}^{T \times d}$      $\triangleright$ observational time-series
  $\tau_{\max}$      $\triangleright$ largest lag to consider
  $\boldsymbol{\epsilon}_\theta(\cdot, k)$      $\triangleright$ A trained diffusion model approximating $\nabla \log p(\mathbf{x})$ at $k \in [0, k_{\max}]$
  $\mathcal{K} = \{k_1, \ldots, k_S\}$      $\triangleright$ selected noise scales
**Ensure:** *orders*      $\triangleright$ List of valid causal orderings
1: **function** $\mathrm{DOTS}(\mathbb{D}, \mathcal{K}, \tau_{\max})$
2:     *orders* $\leftarrow \varnothing$
3:     **for all** $k \in \mathcal{K}$ **do**
4:        $V \leftarrow \{\mathrm{x}_i^\tau \mid i = 1 \ldots d, \ \tau = 0 \ldots \tau_{\max}\}$      $\triangleright$ lag-embedded nodes
5:        $\pi \leftarrow [\,]$      $\triangleright$ ordering for this scale
6:        **while** $V \neq \varnothing$ **do**
7:           $\boldsymbol{H}_{\mathrm{diag}} \leftarrow \mathrm{HESSIANDIAGVAR}\big(\boldsymbol{\epsilon}_\theta, \mathbb{D}[:, V], k\big)$
8:           $L \leftarrow \arg\min_{v \in V} \boldsymbol{H}_{\mathrm{diag}}[v]$      $\triangleright$ leaf(s)
9:           $\pi \leftarrow [L \mid \pi]$
10:          $V \leftarrow V \setminus L$
11:        **end while**
12:        *orders* $\leftarrow$ *orders* $\cup \{\pi\}$
13:     **end for**
14:     **return** *orders*
15: **end function**

---

### 4.4 Aggregating Multiple Orderings

Section 3 showed that taking the *intersection* of *all* topological sorts of a DAG $\mathcal{G}$ yields its transitive closure $G^+$, which is in general a **superset** of $\mathcal{G}$. Because enumerating every ordering is infeasible, we combine a finite sample of orderings in two simple steps.

**Soft voting.** From $S$ orderings $\{\pi_1, \dots, \pi_S\}$ obtained at diffusion steps $k_{1:S}$ we form a vote matrix $W_{ij} = S^{-1} \sum_s \mathbf{1}\{(i \to j) \in \pi_s\}$. Thresholding at $\theta \in (0, 1]$ produces the *soft transitive closure* $\tilde{A}_{ij} = \mathbf{1}\{W_{ij} \geq \theta\}$. The extremes $\theta = 0$ and $\theta = 1$ reduce to the plain union and the hard intersection, respectively; intermediate values let us balance recall against precision.

**CAM pruning.** The matrix $\tilde{A}$ can still contain indirect or spurious edges. We refine it with the likelihood–based pruning routine of Bühlmann et al. (2014), removing edges that do not improve the predictive loss of the child variable given its other parents. The result is our final estimate $\hat{A}$. This procedure is commonly used across most causal ordering approaches from Section 6.1 and constitutes an integral part of our method as it allows us to recover DAG $\mathcal{G}$ from $G^+$ by eliminating indirect links.

**Practical reliability.** Soft voting has the appealing property that any edge appearing in *every* sampled ordering is retained, whereas edges that never appear are discarded automatically. When (i) the sampler generates a diverse set of valid orderings and (ii) the sample size is large enough for CAM's tests to be informative, $\tilde{A}$ approximates $G^+$ increasingly well and the pruning step tends to eliminate the remaining indirect links, often recovering $\mathcal{G}$ exactly in practice.

## 5 Experiments

### 5.1 Setup

Our experimental framework prioritizes replication of results, modularity, and ease of extension. These are features found in the Snakemake (Mölder et al., 2021) workflow management system, that forms a base for our experimental setup. Snakemake has previously been used for benchmarking purposes in (non-temporal) causal discovery (Rios et al., 2025), from which we draw inspiration. Our codebase is accessible online[2].

### 5.2 Data

**Simulations**. Our synthetic Data Generating Process (DGP) is based on the work of Beaumont et al. (2021) and Lawrence et al. (2020). Our experimental setup involves DGPs with the following properties: sample size (observed time steps) $T \in \{200, 1000, 2000, 5000\}$, number of graph nodes $d \in \{3, 4, 5, 6\}$, lag size $\tau \in \{1, 2, 3\}$. In addition, all setups use non-linear causal mechanisms (piecewise linear and trigonometric) and incorporate the same noise distribution $\epsilon \sim \mathcal{N}(0, [0.01, 0.05])$. Each setting has been repeated 10 times to obtain robust results. Causal mechanisms and relationships are invariant across time (i.e. they are stationary).

Borrowing the notation from Lawrence et al. (2020), a set of temporal causal links $\mathcal{T}$ generated in the DGP is defined as follows:

$$\mathcal{T}_{t,\tau} := \{X_i(t - \tau) \to X_j(t) | i, j \in \{1, \dots, d\}\}, \tag{5}$$

with $t$ denoting time index. Note that we do not consider instantaneous links in our experiments ($\tau > 0$), but do allow for autoregressive relationships ($i = j$). We also fix the lag size $\tau$ across all relationships within any single dataset to isolate and study the influence of $\tau$ on algorithmic performance.

**Real datasets**. We also perform experiments on datasets closer to real-life complexities. To achieve this, we incorporate CausalTime (Cheng et al., 2024), a realistic benchmark for time series causal discovery. CausalTime provides three datasets: Air Quality Index (AQI), Traffic, and Medical. The AQI data consist of 36 variables, whereas Traffic and Medical have 20. In terms of sample size, all datasets consist of 480 samples,

---

[2]https://github.com/CHAI-UK/DOTS

with time length of $T{=}40$. Since DOTS requires a large sample size to perform well, we combine all samples (all 480 samples of length $T{=}40$ stacked vertically), plus one row of zeros[3] to avoid cross-contamination, into a single dataset of length $T{=}480 \times (40 + 1) - 1{=}19\,679$. We apply the same pre-processing procedure to all three datasets.

- **AQI**: hourly $PM_{2.5}$ readings from $N{=}36$ monitoring stations across China ($T{=}8760$). A geographical distance kernel supplies a sparse prior graph.

- **Traffic**: average speed measured every 5 min at $N{=}20$ loop detectors in the San-Francisco Bay Area ($T{=}52\,116$); the prior graph again follows pairwise distance.

- **Medical**: $N{=}20$ vital-sign and chart-event channels extracted from 1000 MIMIC-IV ICU stays, resampled to 2-h resolution ($T{=}600$ on average).

### 5.3 Algorithms

We compare our proposed method to the following baselines:

- Temporal:

  - **Dummy**: Returns a fully-connected temporal DAG as a naive estimation.
  - **PCMCI** (Runge, 2020): Employs lagged conditional-independence tests for constraint-based discovery in autocorrelated time series.
  - **PCMCI+** (Runge, 2020): Extends PCMCI with additional conditioning to control false positives.
  - **VARLiNGAM** (Hyvärinen et al., 2010): Combines linear VAR modeling with non-Gaussian ICA to recover a unique causal order.
  - **DYNOTEARS** (Pamfil et al., 2020): Casts temporal DAG learning as a single continuous optimization with an acyclicity constraint.
  - **TCDF** (Nauta et al., 2019): Trains temporal convolutional networks and validates edges via in-silico interventions.
  - **TiMINo** (Peters et al., 2013): Applies additive-noise regressions with independence tests on residuals for both lagged and instant effects.

- Non-temporal:

  - **CAM** (Bühlmann et al., 2014): Fits additive noise models with restricted maximum likelihood and sparsity-based pruning.
  - **SCORE** (Rolland et al., 2022): Uses score-matching to estimate the Hessian variance for iterative leaf removal.
  - **DAS** (Montagna et al., 2023c): Scalable ANM ordering via efficient Hessian diagonal estimation.
  - **NoGAM** (Montagna et al., 2023b): Generalizes ANM ordering without Gaussian noise assumptions, leveraging kernelized score estimates.
  - **DiffAN** (Sanchez et al., 2023): Uses denoising diffusion models to approximate the score Jacobian for fast, retraining-free causal ordering.

While the ordering-based algorithms (CAM, SCORE, DAS, NoGAM, DiffAN) were not developed for temporal tasks, we still include them in our experiments due to our proposed method's strong roots in topological ordering. To make the use of the ordering-based methods more appropriate in this temporal setting, we post-process their predicted graphs by removing the edges that defy the arrow of time. This mild addition is indicated by the '-C' suffix added to the name of the methods in question (e.g. CAM-C) when reporting the results in Section 5.6. Comparing to temporal methods, however, remains the main validation target for us.

---

[3]Separating zeros are not needed at the end of the data, hence minus one in the formula.

### 5.4 Graph Representation

Temporal causal graphs can be represented at different granularities (Assaad et al., 2022):

- **Window Causal Graph:** Restricts the full time causal graph to a finite lag window $\tau_{\max}$, representing only edges from $x_i^{t-\tau}$ to $x_j^t$ for $\tau \le \tau_{\max}$. This trades off completeness for computational feasibility.

- **Summary Causal Graph:** Aggregates causal relationships across time without specifying exact lag indices, creating a more compact but less detailed representation as lagged and contemporaneous edges are represented the same way. Autocorrelated variables are represented with self-loops.

Window graphs are naturally more suitable to our method's use case. However, we also include summary graphs in our study as we build upon TiMINo that outputs only this type of graphs. Including summary graphs allows us to directly compare to TiMINo.

### 5.5 Evaluation

The assumption that SCMs are invariant across time (stationarity) results in repeated causal links. Therefore, we focus on the correctness of the predicted edges that terminate at (non-lagged) time $t$, that is $\mathcal{T}_{t,\tau}$. We then compare predicted edges to the ground truth and calculate True Positives (TP), False Positives (FP) and False Negatives (FN), from which we obtain *Recall*, *Precision* and $F1$ metrics as follows:

$$\text{Recall} = \frac{TP}{TP+FN}, \qquad \text{Precision} = \frac{TP}{TP+FP}, \qquad F1 = \frac{2 \times \text{Recall} \times \text{Precision}}{\text{Recall} + \text{Precision}}, \qquad (6)$$

as per (Assaad et al., 2022). We report $F1$ on window and summary graphs ($F1_W$ and $F1_S$, respectively) as the main metric of interest, but we also supplement our results with *Recall* and *Precision*.

### 5.6 Results

#### 5.6.1 Simulations

Figures 6 and 7 show the main results (averages and 95% confidence intervals). In both cases (window and summary graphs), DOTS shows very strong and robust performance **across different sample sizes, numbers of features and lag sizes**. Apart from a clear separation from the competition, DOTS is also one of the few methods that keeps improving in larger sample sizes ($T = 5000$). Ordering-based methods and *PCMCI* family perform comparatively, with another diffusion-based method (*DiffAN-C*) coming out on top among these, showing the advantage of diffusion models in strongly nonlinear tasks. *VARLiNGAM*, *DYNOTEARS* and *TCDF* struggle to outperform the naive baseline that predicts fully-connected DAGs, suggesting their leniency towards linear settings.

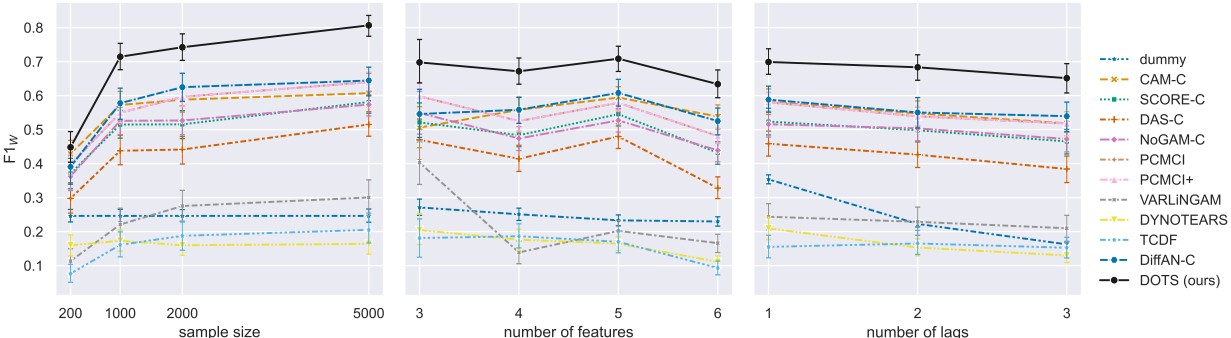

Figure 6: F1 scores on simulated window graphs ($F1_W$). TiMINo does not provide $F1_W$.

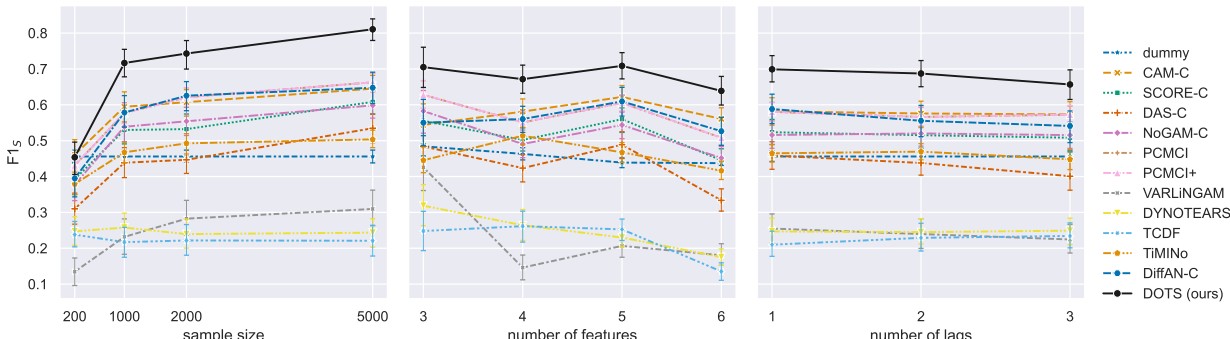

Figure 7: F1 scores on simulated summary graphs ($F1_S$).

Figure 8 shows average running times of all methods across the simulations. DOTS places in the middle among the competitors, providing strong prediction performance at no extra computational costs as compared to baselines. The runtime of DOTS also scales well that shows its promise in high-dimensional tasks.

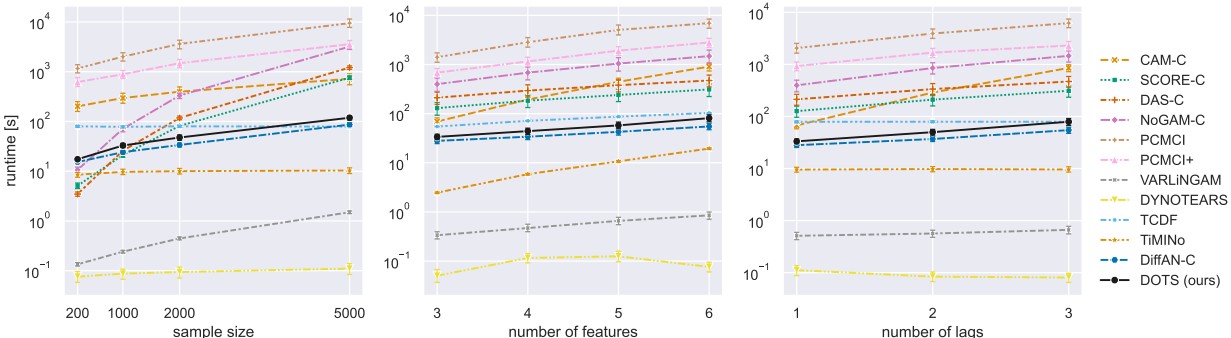

Figure 8: Average runtime of each algorithm in seconds obtained on synthetic data. Note the logarithmic scale on the y axis.

### 5.6.2 Real Datasets

Table 2 summarises the main findings. Two broad trends emerge.

1. **Overall difficulty.** Average $F1_S$ rarely exceeds 0.45, far below synthetic baselines, underscoring the gap between toy DGPs and real dynamics. Constraint-based *PCMCI* loses precision on the noisier Medical subset, while score-based *DYNOTEARS* collapses on AQI potentially due to its linearity assumption. On average, however, many temporal methods are competitive except for *TCDF* and *DYNOTEARS*.

2. **Best-in-class methods.** Among approaches with functional assumptions, *TiMINo*, which leverages the same function assumptions as DOTS, excels on the Medical data. *VARLiNGAM* dominates the spatial datasets, hinting that weak non-Gaussianities suffice for identifiability when the linear VAR fit is adequate, though both TiMINo and PCMCI are not far behind. Our method (DOTS), despite not achieving the top rank on individual datasets, is within one standard error of the leader on every subset and achieves the highest aggregate mean, validating the multi-ordering strategy.

Qualitatively, failures tend to cluster around long-range edges (large spatial distance in AQI/Traffic or cross-system interactions in Medical). Future work should explore explicit distance-aware regularisation and non-stationary mechanisms to close this performance gap.

Table 2: $F1_S$ (higher is better) on CausalTime. Best and second best per column are highlighted. SCORE, DAS and NoGAM (non-temporal) have been excluded due to exceeded memory allocation limits.

| Type | Method | Medical | AQI | Traffic | Avg. ↑ |
|---|---|---|---|---|---|
| non-temporal | CAM-C | 0.413 | 0.351 | 0.304 | 0.356 |
| | DiffAN-C | 0.313 | 0.356 | 0.282 | 0.317 |
| temporal | PCMCI/PCMCI+ | 0.427 | 0.382 | 0.372 | 0.393 |
| | VARLiNGAM | 0.457 | **0.464** | **0.391** | 0.437 |
| | DYNOTEARS | 0.107 | 0.010 | 0.315 | 0.144 |
| | TCDF | 0.286 | 0.218 | 0.000 | 0.168 |
| | TiMINo | **0.553** | 0.429 | 0.340 | 0.441 |
| | DOTS (ours) | 0.548 | 0.457 | 0.349 | **0.451** |

### 5.6.3 Ablation on the number of orderings

We now study how the number of causal orderings influences the predictive performance of DOTS. Our theoretical results in Section 3 consider the case where *all* valid causal orderings are present. In Figure 3, we study the impact of the number of orderings in an ideal scenario where *all* causal orderings are known. DOTS generates a small, finite set of orderings based on a heuristic: using different diffusion timesteps ($k$). Therefore, we run this experiment to validate that, indeed, orderings derived from different $k$ are effective in improving performance.

To investigate this, we run DOTS on synthetic data ($T = 5000$) generated from a 3-node graph with $\tau = 1$. We repeat the experiment 500 times. The results are shown in Figure 9. Sampling a single causal ordering is underperforming as more instances are needed to arrive at the right graph. On the other hand, as few as four orderings show a substantial performance improvement, which is on the same performance level as ten orderings. Moving further to 15 instances shows a mild improvement, beyond which the performance plateaus. Overall, selecting the number of sampled causal orderings clearly has a large effect on method predictive performance, and while exploring more orderings may guarantee better performance, staying in the range of 4–10 may often strike the right balance between computational cost and delivered performance. The asymptotic gains are in agreement with the theoretical results presented in Figure 3.

*False positives* (spurious edges) are pruned more effectively because conflicting orderings rarely vote for the same incorrect link, while *false negatives* (missed true edges) are rescued when at least one ordering captures the correct ancestor–descendant relation. These findings empirically substantiate our theoretical claim from Section 3: aggregating multiple orderings provides a robust consensus estimate of the transitive closure, correcting errors that any single ordering may introduce.

We also study the influence of the *max lag* $\tau_{\max}$ and soft-voting threshold $\theta$ hyperparameters on performance of DOTS, as well as the diversity of orderings obtained at different noise scales $k$. The results for these can be found in Appendix C.

### 5.7 Limitations

We would like to highlight that the robustness of assumption violations tested in Montagna et al. (2023a), such as faithfulness, sufficiency, functional, and others, are not covered in our empirical evaluation. Instead, assuming stationarity and causal sufficiency, our robustness checks investigate variations in sample size, number of variables and size of lagged relationships. As a result, any robustness claims we make throughout this work pertain specifically to those data characteristics we tested, and not the ones investigated by Montagna et al. (2023a). Future work should investigate our method's robustness to non-stationary and confounded settings.

Another important consideration is that DOTS relies on CAM pruning and hence inherits its limitations. However, by first recovering the theoretically grounded transitive closure, we provide a stronger foundation for subsequent pruning than single-ordering methods can achieve.

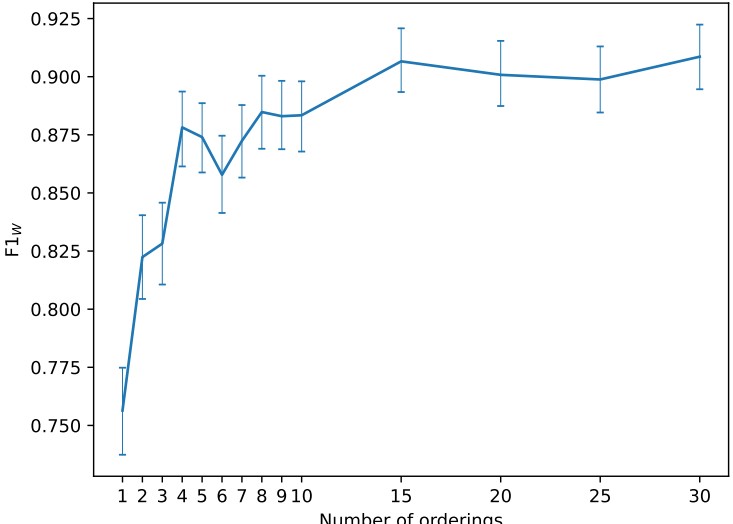

Figure 9: The relationship between prediction performance ($F1_W$) and the number of explored causal orderings in DOTS.

# 6 Related work

## 6.1 Ordering-Based Causal Discovery

Representing a DAG by one of its valid topological orderings (Verma & Pearl, 1990) reduces structure learning to a permutation search followed by edge selection. Early work framed this idea as a discrete optimisation problem: greedy MCMC over permutations (Friedman & Koller, 2003), hill-climbing with dynamic programming (Teyssier & Koller, 2005), arc-reversal searches (Park & Klabjan, 2017), and restricted-MLE procedures such as CAM (Bühlmann et al., 2014). More recent combinatorial schemes enforce sparsity via $\ell_0$-penalised likelihoods, yielding the "sparsest permutation" estimators with provable consistency guarantees (Raskutti & Uhler, 2018; Solus et al., 2021; Lam et al., 2022). Reinforcement-learning formulations further cast the permutation search as a sequential decision process, amortising exploration across datasets (Wang et al., 2021).

Beyond score optimisation, identifiability can be strengthened by exploiting distributional or algebraic asymmetries. In linear additive models, sequentially peeling off leaf nodes from the precision matrix recovers the causal order under heteroscedastic noise assumptions (Ghoshal & Honorio, 2018; Chen et al., 2019). Deterministic functional constraints are handled by Determinism-aware GES (DGES), which first clusters exact relations and then performs exact search within each cluster, using determinism itself as an unambiguous ordering cue (Li et al., 2024). Non-Gaussianity offers an alternative route: LiNGAM identifies a unique ordering via independent-component analysis, a principle extended to functional data in Func-LiNGAM (Shimizu et al., 2006).

Scalability to high-dimensional, nonlinear settings has recently been advanced through continuous relaxations and deep generative models. CaPS estimates Hessian diagonals of the log-likelihood to iteratively detect leaves, unifying linear and nonlinear mechanisms and accelerating pruning with a "parent score" metric (Xu et al., 2024). Recent advances in ordering-based causal discovery further refine and generalise score-based estimation strategies. SCORE (Rolland et al., 2022) leverages score matching techniques to iteratively identify and remove leaf nodes, specifically utilizing variance estimates of the Hessian diagonal. Building upon similar principles, DAS (Montagna et al., 2023c) enhances scalability by efficiently estimating Hessian

diagonals, significantly reducing computational overhead. DiffAN trains a denoising diffusion model to approximate the score-function Jacobian, introducing a deciduous update rule that circumvents network retraining during iterative leaf removal, thus scaling ordering discovery to hundreds of variables (Sanchez et al., 2023).

More recently, there have been notable efforts to generalise the score-matching framework. For instance, NoGAM (Montagna et al., 2023b) generalises ordering-based approaches beyond Gaussian noise assumptions, employing kernelised score estimates to accommodate a wider range of data distributions. Furthermore, Liu et al. (2024) relaxes the assumption that all models must be either linear or nonlinear and considers problems with mixed models, notably also leveraging parallel processing to improve scalability.

Together, these developments illustrate a shift from discrete combinatorics to differentiable optimisation, while preserving the core insight that a well-chosen causal ordering sharply narrows the search for a faithful DAG. None of these works explore combining multiple valid causal orderings to recover the full adjacency matrix.

## 6.2 Hessian of the Log-likelihood

Estimating $\boldsymbol{H}(\log p(\mathbf{x}))$ is the most expensive task of the ordering algorithm. Our baseline (Rolland et al., 2022) proposes an extension of Li & Turner (2018) which utilises Stein's identity over an RBF kernel (Schölkopf & Smola, 2002). Rolland et al.'s method cannot obtain gradient estimates at positions out of the training samples. Therefore, evaluating the Hessian over a subsample of the training dataset is impossible. Other promising kernel-based approaches rely on spectral decomposition to solve this (Shi et al., 2018) issue and constitute promising future directions. Most importantly, computing the kernel matrix is expensive for memory and computation on $n$. There are, however, methods (Achlioptas et al., 2001; Halko et al., 2011; Si et al., 2017) that help to scale kernel techniques not considered in the present work. Other approaches are also possible with deep likelihood methods such as normalising flows (Durkan et al., 2019; Dinh et al., 2016) and further computing the Hessian via backpropagation. This would require two backpropagation passes giving $O(d^2)$ complexity and be less scalable than denoising diffusion. Indeed, preliminary experiments proved impractical in our high-dimensional settings.

We use DPMs because they can efficiently approximate the Hessian with a single backpropagation pass while allowing Hessian evaluation on a subsample of the training dataset. It has been shown (Song & Ermon, 2019) that denoising diffusion can better capture the score than simple denoising (Vincent, 2011) because noise at multiple scales explores regions of low data density.

## 6.3 Causal Discovery for Time Series

Granger's seminal definition of causality for time series—past $X$ improves the prediction of future $Y$—still underpins most modern approaches (Granger, 1969). Structural causal models (SCMs) extend this idea to permit intervention semantics, latent variables, and cycles (Bongers et al., 2021). Constraint-based algorithms such as PCMCI and its refinement PCMCI+ adapt the PC procedure to lagged conditional-independence testing, enabling scalable false-discovery control in autocorrelated, high-dimensional settings (Runge, 2020). Functional-form assumptions provide stronger identifiability: TiMINo employs additive-noise regressions with independence tests on residuals (Peters et al., 2013), while VARLiNGAM couples a linear VAR with non-Gaussian errors and independent-component analysis to recover causal ordering (Hyvärinen et al., 2010).

Score-based and deep-learning methods further relax linearity and stationarity. DYNOTEARS casts structure learning as a single differentiable optimisation problem over lagged and contemporaneous edges with an acyclicity constraint (Pamfil et al., 2020). TCDF trains attention-based CNNs and validates candidates through *in-silico* interventions (Nauta et al., 2019), whereas CausalFormer augments Transformers with causality-aware attention to handle long sequences (Kong & Lu, 2024). Continuous-time dynamics can now be unveiled with sparse Neural ODEs that yield interpretable differential systems from irregular samples (Aliee et al., 2023). Information-theoretic criteria such as transfer entropy generalise Granger tests to non-linear interactions, though density estimation remains costly in high dimensions (Schreiber, 2000). Most

recently, PICK introduces score-matching algorithms to temporal causal discovery while notably reducing pruning time by exploiting the variance of the scores, offering an alternative path to efficiency (Chen et al., 2024). Recent surveys also emphasise persistent challenges—hidden confounders, non-stationarity, and computation at scale (Gong et al., 2023; Assaad et al., 2022; Moraffah et al., 2021). Our method advances the field by combining the statistical rigour of functional approaches with the scalability of continuous optimisation while accommodating nonlinearities typical of real-world data.

## 7 Conclusion

In this work, we introduced DOTS, a diffusion-based approach leveraging multiple causal orderings to address the challenge of temporal causal discovery. While previous single-ordering methods were primarily developed in the context of static causal discovery, our work extends the causal ordering framework explicitly to the temporal setting. This temporal context inherently incorporates the causal temporality principle, where variables can only causally influence future variables, not past ones. Unlike traditional single-ordering methods, DOTS effectively captures complementary information by aggregating multiple valid causal orderings, thereby reconstructing the transitive closure of the underlying temporal DAG. We formalized the theoretical benefits of this multi-ordering strategy, demonstrating its capacity to mitigate spurious dependencies and enhance robustness in causal inference. Our empirical results on synthetic datasets clearly illustrate the superiority of DOTS compared to existing state-of-the-art baselines in terms of accuracy and scalability, while staying competitive and the best on average on real data. By exploiting the inherent frequency domain characteristics of diffusion steps, our method provides nuanced insights into both coarse and fine-grained temporal causal interactions. Future research directions include exploring additional aggregation strategies for causal orderings, extending our method to non-stationary environments, and further optimizing computational efficiency for large-scale applications.

### Acknowledgments

We would like to thank Ricardo Silva and anonymous reviewers for valuable comments and suggestions. We acknowledge the support of the UKRI AI programme, and the Engineering and Physical Sciences Research Council (EPSRC), for the Causality in Healthcare AI Hub [grant number EP/Y028856/1].

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

# A    Order theory definitions

Here, we establish the foundational definitions used in order theory.

**Definition 1** (Partial Order). Let $V$ be a set. A binary relation $\leq$ on $V$ is a *partial order* if, for all $x, y, z \in V$, it satisfies:

    (i) **Reflexivity**: $x \leq x$.

    (ii) **Antisymmetry**: If $x \leq y$ and $y \leq x$, then $x = y$.

    (iii) **Transitivity**: If $x \leq y$ and $y \leq z$, then $x \leq z$.

The pair $(V, \leq)$ is called a *partially ordered set* (or *poset*).

**Definition 2** (Strict Partial Order). A binary relation $\prec$ on $V$ is a *strict partial order* if it is:

    (i) **Irreflexive**: For all $x \in V$, $x \nprec x$.

    (ii) **Transitive**: If $x \prec y$ and $y \prec z$, then $x \prec z$.

**Definition 3** (Linear Extension). For a poset $(V, \leq)$, a *linear extension* is a total order $\preceq$ on $V$ such that if $x \leq y$, then $x \preceq y$ for all $x, y \in V$. That is, $\preceq$ extends $\leq$ into a total ordering consistent with the partial order.

**Definition 4** (Reachability Relation). For a directed graph $G = (V, E)$, the *reachability relation* $R \subseteq V \times V$ is defined as:

$$(x, y) \in R \iff \text{there exists a directed path from } x \text{ to } y \text{ in } G.$$

If $G$ is a DAG, $R$ is a partial order on $V$ (reflexive, antisymmetric, and transitive), as every vertex is reachable from itself via a trivial path.

**Definition 5** (Transitive Closure). For a directed graph $G = (V, E)$ with reachability relation $R$, the *transitive closure* of $G$ is the graph:

$$G^+ = (V, E^+),$$

where:

$$E^+ = \{(x, y) \in V \times V : (x, y) \in R \text{ and } x \neq y\}.$$

Thus, $\mathcal{G}^+$ contains an edge $(x, y)$ if and only if there is a non-trivial directed path from $x$ to $y$ in $G$, excluding self-loops.

# B  Supplementary Results

## B.1  Alternative metrics

Figures 10 and 11 supplement our main simulation results. In these, we report precision and recall on both window and summary graphs. The most important observation here is that the top-performing methods almost never provide non-existent edges (i.e. high precision), but some undetected edges still remain (i.e. recall lower than 1). Future work could focus on improving edge detection while maintaining high precision levels.

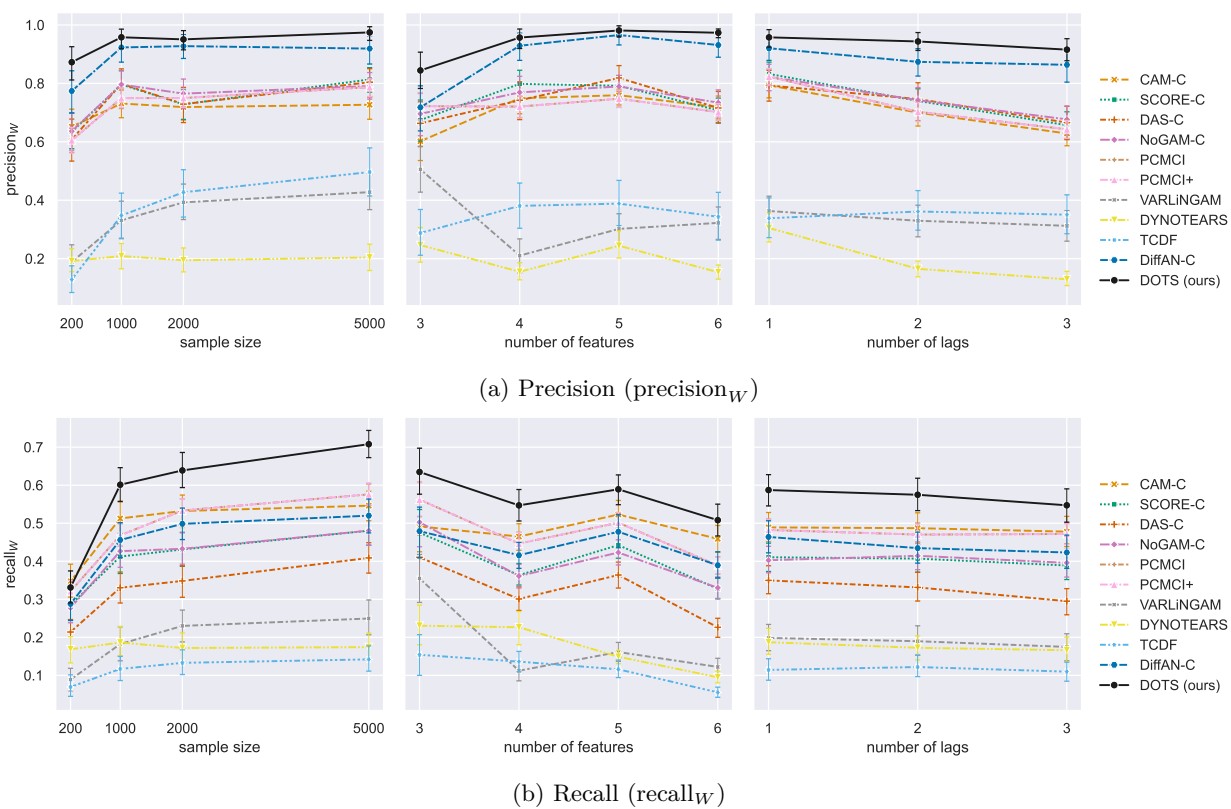

(a) Precision ($\text{precision}_W$)

(b) Recall ($\text{recall}_W$)

Figure 10: Precision and recall (higher is better) on simulated **window** graphs. TiMINo does not provide window graphs.

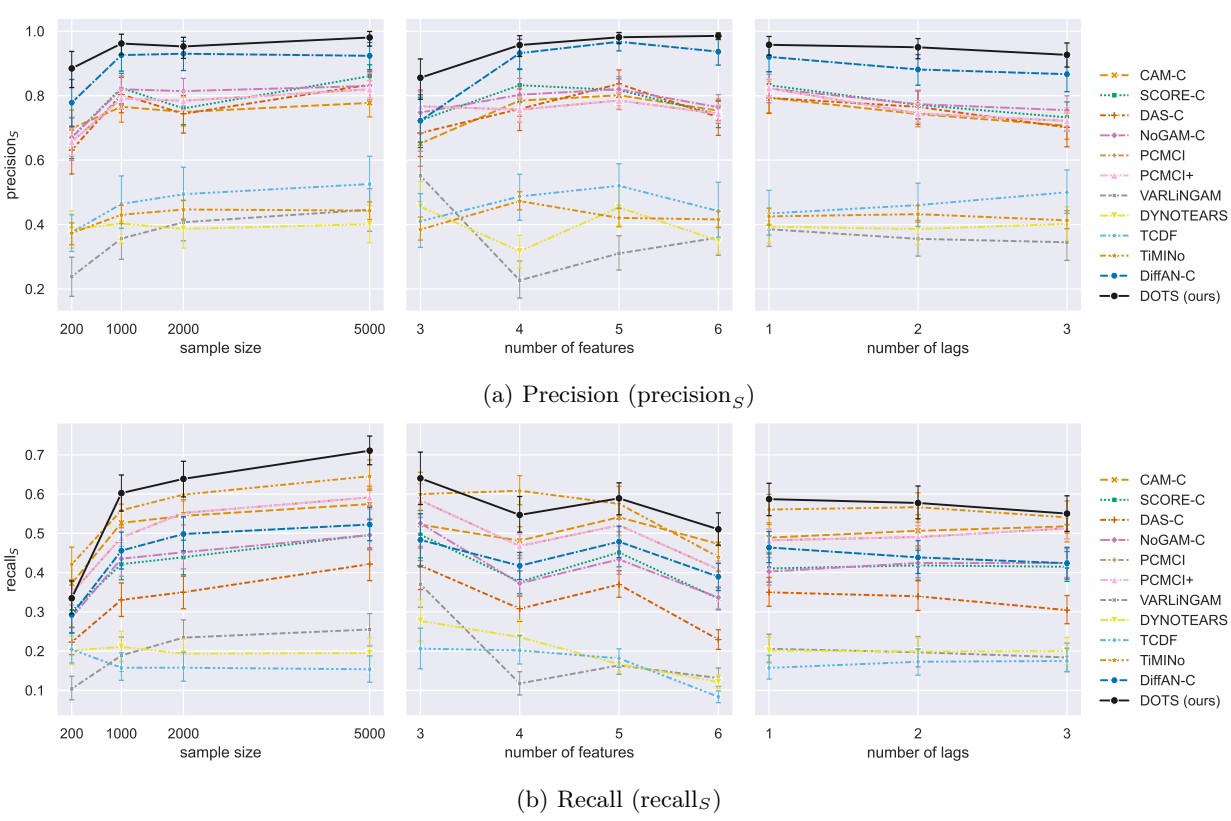

(a) Precision ($\text{precision}_S$)

(b) Recall ($\text{recall}_S$)

Figure 11: Precision and recall (higher is better) on simulated **summary** graphs.

## B.2  Effect of the temporal constraint

Figure 12 shows the influence of the temporal constraint applied to ordering-based methods to ensure they return a graph that respects the arrow of time. Most methods show some mild performance improvement and mostly in higher precision. Interestingly, diffusion-based methods (*DiffAN* and DOTS) do not exhibit such a benefit, which could be partly due to already high precision achieved even without the constraint (i.e. not much room for further improvement).

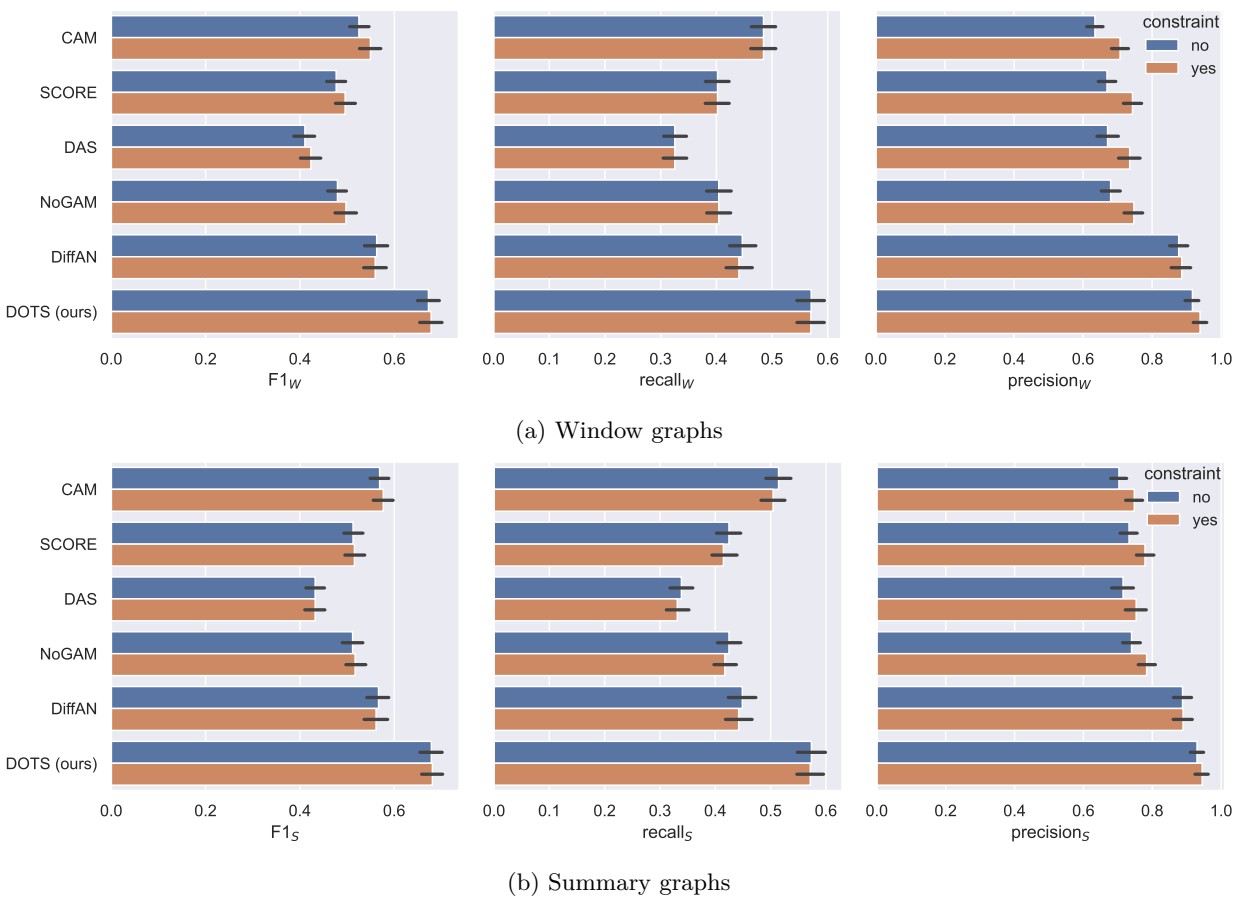

(a) Window graphs

(b) Summary graphs

Figure 12: The effect of the temporal constraint on the performance of score-matching algorithms.

# C Ablations

## C.1 Maximum lag

DOTS has a hyperparameter that sets the maximum lag ($\tau_{\max}$) to be considered in the predicted graph. Here we study its influence on predictive performance. To this end, we run simulations with 3-node graphs, $T{=}5000$ and $\tau \in \{1, 2, 3\}$, all repeated 100 times. Figure 13 presents the results.

Two main observations emerge. First, selecting $\tau_{\max}$ shorter than ground truth can have an obvious significant negative impact on performance as it prevents DOTS from identifying the necessary longer relationships. However, setting $\tau_{\max}$ to values larger than the truth maintains good performance as compared to when the hyperparameter matches the lag size in the data. This has important practical implications, suggesting that setting $\tau_{\max}$ to larger values may provide more robust results since the right lag size is unknown when processing real data.

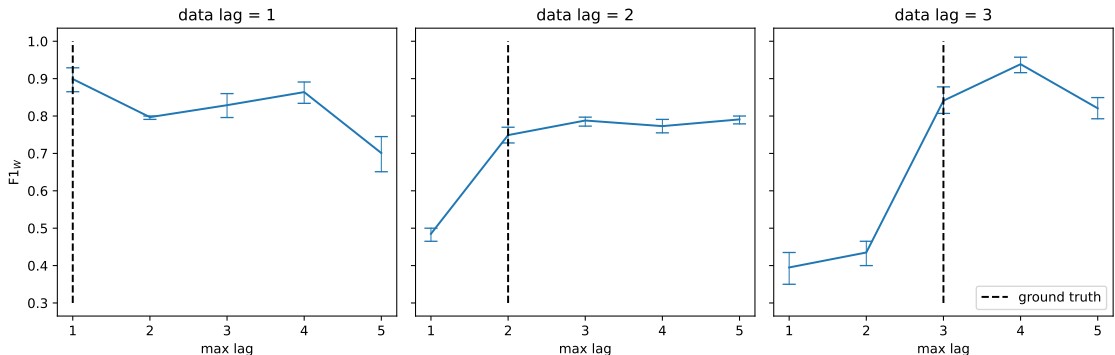

Figure 13: The relationship between prediction performance ($F1_W$) and the maximum lag (*max lag*) hyperparameter in DOTS. Each setting (subfigure) includes one type of lag in the data (*data lag*), which is also highlighted by the vertical *ground truth* line. Error bars denote 95% confidence intervals.

## C.2 Soft-voting threshold

Here we investigate the sensitivity of DOTS to the soft-voting hyperparameter $\theta$. We run simulations with 3-node graphs, $T$=5000, $\tau = 1$, all repeated 100 times. Figure 14 presents the results.

The main observation is a clear trend, in which performance consistently degrades as we increase the threshold. This is consistent with the behaviour we expect as increasing the threshold requires more votes from each ordering for a graph edge to be present, which inevitably leads to increased false negatives and decreased recall (middle subfigure).

Interestingly, union of all orderings ($\theta = 0$) provides the best results, and which is notably free from false positives (i.e. perfect precision as in the right-hand-side subfigure). This result shows importance and effectiveness of CAM pruning since without it the number of false positives would likely be very high. This observation also makes $\theta = 0$ the recommended default value.

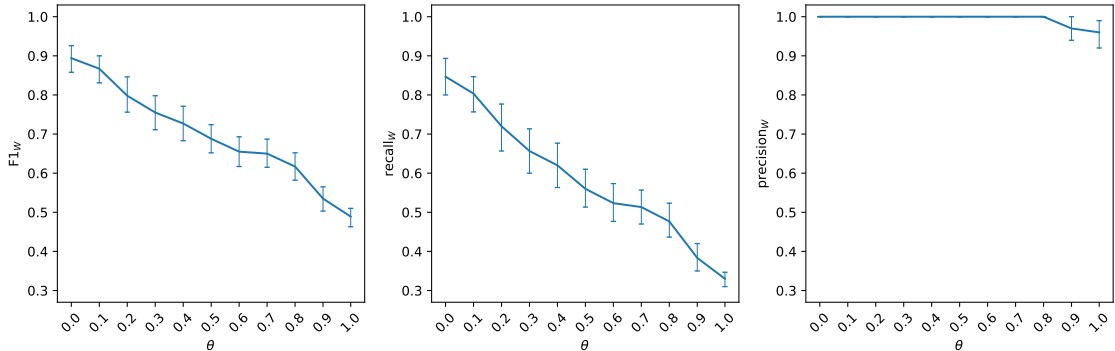

Figure 14: The relationship between prediction performance and the soft-voting threshold $\theta$ hyperparameter in DOTS. Error bars denote 95% confidence intervals.

### C.3 Diversity of orderings

To quantify the diversity of orderings produced during a training run, we computed pairwise distances between all permutations using the Kendall tau metric. This distance reflects the proportion of pairwise disagreements in item ordering between two permutations and is widely used for comparing ranked lists. By converting each permutation into its corresponding rank vector and computing normalized Kendall distances, we obtain a symmetric distance matrix that captures how similar or dissimilar each ordering is to every other within the same run.

Figure 15 visualises this matrix as a heatmap. Blocks of darker regions indicate sets of orderings that remain relatively consistent, while lighter bands correspond to greater variability across orderings. The structured patterns suggest that the model explores multiple, partially overlapping ranking modes rather than collapsing to a single consensus. In particular, similar k values yield similar orderings as shown by the block patterns on the diagonals.

In Figure 16 we further investigate predictive performance of individual orderings at different noise scale values $k$. Each ordering was passed through CAM pruning to get the final performance measure against the true DAG. The figure implies that different noise scales provide different edge-detection capabilities, especially in the $[0, 30]$ range. Importantly, however, none of the orderings achieves individually as high of a performance as when all orderings at different noise scales are combined into a single robust prediction.

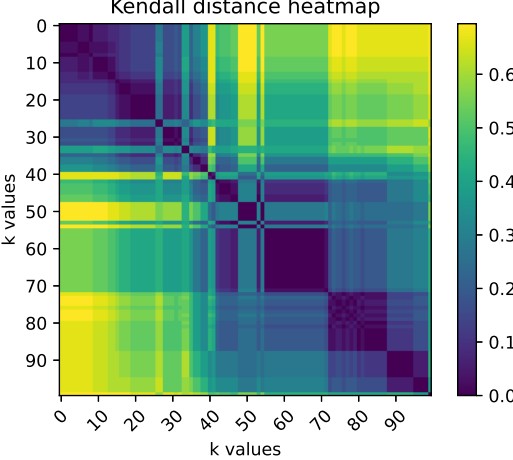

Figure 15: Kendal distance between different orderings obtained at different noise scale values $k$.

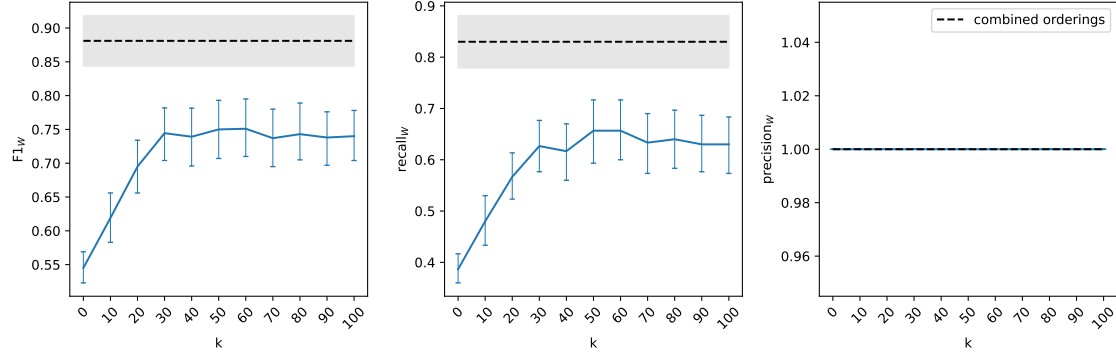

Figure 16: Performance of individual orderings obtained at different noise scales $k$ compared to the performance achieved with combined orderings. Results are averages over 100 repetitions. Error bars and shaded grey areas denote 95% confidence intervals.

# D  Experimental Details

## D.1  Hyperparameters

We summarise the most important hyperparameters used in conjunction with the methods in Table 3. For the rest of the hyperparameters not mentioned here, we deferred to their default values, which can be found in their respective implementations as per Table 4.

In addition, most methods have a hyperparameter that corresponds to the maximum lag size, though can be named differently in their implementations ($\tau_{max}$ in PCMCI/PCMCI+, *lags* in VARLiNGAM, *p* in DYNOTEARS, *max_lag* in TiMINo). This hyperparameter can have a non-trivial effect on method's performance. In order to decrease our experiments' dependence on hyperparameter tuning, we set this hyperparameter to the true lag value $\tau$ in the simulations since we have access to the ground truth. In real data settings (CausalTime), we set the value to 1. Note that some methods (CAM, SCORE, DAS, No-GAM, DiffAN, DOTS) implement this hyperparameter implicitly by creating $\tau \times d$ lagged variables based on provided data.

While sampling 10 causal orderings in DOTS is a reasonable default (n_ord), we increase it to 20 when processing CausalTime to better showcase our method's potential.

Table 3: Summary of hyperparameters of all methods used in the experiments.

| Method | Hyperparameters |
|---|:---:|
| CAM | alpha $= 0.05$ |
| SCORE | $\alpha = 0.05, \eta_G = 0.001, \eta_H = 0.001$ |
| DAS | $\alpha = 0.05, \eta_G = 0.001, \eta_H = 0.001$ |
| NoGAM | $\alpha = 0.05, \eta_G = 0.001, \eta_H = 0.001$ |
| | $\text{ridge}_\alpha = 0.01, \text{ridge}_\gamma = 0.1$ |
| PCMCI/PCMCI+ | $\alpha = 0.05, \text{test} \in \{\text{par\_corr, cmi\_knn}\}, \tau_{min} = 1$ |
| VARLiNGAM | $\alpha = 0.05, \text{criterion} = \text{bic}, \text{prune} = \text{true}$ |
| DYNOTEARS | $\lambda_w = 0.05, \lambda_a = 0.05, \text{w\_threshold} = 0.01$ |
| TCDF | $\text{epochs} = 5000, \text{layers} = 2, \text{lr} = 0.01$ |
| | $\text{kernel\_size} = 4, \text{dilation} = 4, \text{significance} = 0.8$ |
| DiffAN | $\text{steps} = 100, \text{nn\_depth} = 3, \text{batch\_size} = 1024$ |
| | $\text{early\_stop} = 300, \text{lr} = 0.001$ |
| TiMINo | $\alpha = 0.05$ |
| DOTS | $\text{steps} = 100, \text{nn\_depth} = 3, \text{batch\_size} = 1024$ |
| | $\text{early\_stop} = 300, \text{lr} = 0.001, \text{n\_ord} = 10$ |

## D.2  Implementations of algorithms

Table 4: Summary of source code used to run the methods in the experiments.

| Method | Source |
|---|:---:|
| CAM/SCORE/DAS/NoGAM | dodiscover: https://github.com/py-why/dodiscover |
| PCMCI/PCMCI+ | tigramite: https://github.com/jakobrunge/tigramite |
| VARLiNGAM | lingam: https://github.com/cdt15/lingam |
| DYNOTEARS | causalnex: https://github.com/mckinsey/causalnex |
| TCDF | https://github.com/M-Nauta/TCDF |
| DiffAN | https://github.com/vios-s/DiffAN |
| TiMINo | https://github.com/ckassaad/causal_discovery_for_time_series/blob/master/baselines/scripts_R/timino.R |

