# OpenReview forum: "Causal Ordering for Structure Learning from Time Series"
_TMLR — Accepted by TMLR_

### Review · Reviewer_t6jd · 2025-08-25

**Summary Of Contributions:**

The paper introduces DOTS, a method for causal discovery in time series data. The core idea is to move beyond ordering-based methods that rely on a single causal ordering. Instead, DOTS generates multiple causal orderings and aggregates them to improve robustness of the discovered graph. The authors claim that this aggregation process effectively recovers the transitive closure of the true underlying DAG, which helps filter out spurious edges that often arise from single-ordering approaches.

To generate a diverse set of orderings, the method uses a denoising diffusion model, where different noise scales are used to produce distinct orderings. The authors provide a theoretical argument that aggregating all valid topological orderings of a DAG recovers its transitive closure. Empirically, the paper shows that DOTS outperforms several methods on both synthetic and real-world time series datasets.

**Audience:**

Yes

**Broader Impact Concerns:**

I don't have any concerns on the ethical implications of the work that would require adding a Broader Impact Statement.

**Claims And Evidence:**

Yes

**Requested Changes:**

1.  The claims regarding the novelty of using multiple orderings might need to be toned down. The work can be framed as an effective ensemble-based method for an existing ordering-based algorithm (DiffAN).

2.  The paper can include a discussion on the goal of recovering the transitive closure ($\mathcal{G}^{+}$). The authors might want to explicitly state that the ultimate goal is the true DAG ($\mathcal{G}$) and that $\mathcal{G}^{+}$ contains indirect relationships, which can be a limitation. The role of the final pruning step in moving from $\mathcal{G}^{+}$ to an estimate of $\mathcal{G}$ should be made clearer and more central to the method's description.

3.  The link between different diffusion noise scales (k) and the generation of diverse, complementary causal orderings must be substantiated. This requires more than the current high-level intuition. An analysis measuring the diversity of the permutations generated at different scales and linking this diversity to performance would be useful.

4.  The experimental section can be revised to focus on fairer comparisons. The primary comparison should be against methods explicitly designed for time series. The results from adapted static methods should be presented with the clear caveat that they are not designed for this setting. Furthermore, the discussion of real-world results needs to acknowledge datasets where baselines like TiMINo and VARLINGAM outperform DOTS.


Minor changes

- The paper would be improved by a clearer justification for using a diffusion model to estimate the score's Hessian over other potential methods (e.g., normalizing flows). A brief discussion of the specific advantages (scalability, stability, the multi-scale property) would strengthen the motivation.

- The ablation study in Figure 9 is good, but it could be improved. I recommend adding a comparison between the aggregated result and the performance of the best single ordering chosen from the generated set.

**Strengths And Weaknesses:**

Strengths

1. The proposed method, DOTS, demonstrates good performance on synthetic benchmarks, consistently outperforming the baselines. It also achieves competitive, state-of-the-art results on the real-world CausalTime benchmark.

2. The central idea of aggregating multiple causal orderings to improve robustness is intuitive and well-motivated. This ensemble-like approach proves to be highly effective in practice, as demonstrated by the strong results and the ablation study (Fig. 9).

3. The paper presents a creative application of diffusion models for a causal discovery task. Using different noise scales as a heuristic to generate a diverse set of causal orderings is a novel mechanism that connects generative modeling with structural learning.

4. The experimental setup is comprehensive. The authors validate their method across synthetic datasets with varying sample sizes, feature dimensions, and lag structures, as well as on multiple real-world datasets.



Major Weaknesses

1. The central claim is that leveraging multiple orderings is a novel contribution. However, aggregating the output of a model run with different initializations or hyperparameters is a standard ensemble technique. The paper frames this as a fundamental insight into causal discovery, but it can also be seen as an application of ensembling to an existing ordering-based discovery algorithm (DiffAN).

2. The paper argues that recovering the transitive closure $\mathcal{G}^{+}$ is a desirable outcome. The goal of causal discovery is to find the true causal graph $\mathcal{G}$, which represents direct causal relationships. The transitive closure $\mathcal{G}^{+}$  includes all indirect causal pathways, which can be misleading. For example, if A→B→C, the transitive closure includes the edge A→C, which is an indirect effect. In many scientific applications, distinguishing direct from indirect causes is important. While the authors use a final pruning step (CAM pruning) to remove indirect edges, the core theoretical argument and the aggregation method are aimed at recovering $\mathcal{G}^{+}$.

3. The method for generating different orderings relies on running the leaf-finding algorithm at different noise scales k of the diffusion model. The paper provides a high-level intuition that different scales capture different frequency components of the data. However, this link can be more rigorously established. Are the orderings generated from different k values truly diverse and complementary? Or are they minor variations of each other? The ablation study in Figure 9 shows that performance increases with the number of orderings, but this could be due to getting more "shots" at a good ordering.

4. Several of the baselines (CAM, SCORE, DAS, NOGAM, DiffAN) are methods for static data, which the authors adapt to the temporal setting by applying a post-hoc filter to remove edges that violate temporal precedence ("-C" suffix). These methods were not designed for time series and may perform poorly due to their inability to properly handle autocorrelation and other temporal dependencies. The strong performance of DOTS relative to these "constrained" baselines is therefore not surprising.

5. While the paper includes strong time-series baselines like PCMCI+ and TiMINo, the overall narrative is built around outperforming adapted static methods. On the real-world CausalTime benchmark, the results are much closer. DOTS achieves the highest average F1, but TiMINo is better on the Medical dataset and VARLINGAM is better on AQI and Traffic.


Minor Weaknesses

- The paper could better motivate why a diffusion model is necessary here. The core machinery for finding leaves is the Hessian of the log-likelihood. The diffusion model is used as one way to estimate this Hessian. The authors should clarify if other Hessian estimation methods could be used and what the specific advantages of the diffusion-based approach are.

- The ablation in Figure 9 is useful, but it would be strengthened by also showing the performance of individual orderings sampled from different k.

---

> ### Author Response · Authors · 2025-09-11
> **Response to Reviewer t6jd (1/2)**
>
> **Strengths acknowledged**: We appreciate your recognition of our strong empirical performance ("good performance on synthetic benchmarks," "competitive, state-of-the-art results"), the intuitive nature of our approach ("central idea...is intuitive and well-motivated"), and our creative technical contribution ("creative application of diffusion models," "novel mechanism").
>
> We address your concerns systematically:
>
> ### Major Weakness 1: Novelty Claims vs. Standard Ensembling
>
> **The concern**: Multiple orderings can be seen as standard ensembling applied to DiffAN rather than a fundamental causal discovery insight.
>
> **Our response**: This is a fair reframing suggestion. We acknowledge that at a high level, aggregation resembles ensembling.
>
> **Key distinctions in our contribution**:
>
> - **Theoretical foundation**: Our Proposition 1 provides formal justification that aggregating **all** topological orderings recovers the transitive closure $G^+$, not just empirical ensemble benefits.
>
> - **Novel generation mechanism**: Using diffusion noise scales to systematically generate diverse orderings is distinct from random initialization ensembling.
>
> - **Temporal integration**: The combination with temporal constraints creates structure learning advantages beyond generic ensembling.
>
> **Manuscript changes**:
>
> - **Reframed positioning**: Updated abstract/introduction to describe our approach as "theoretically motivated ensemble-based method" rather than claiming fundamental novelty of aggregation per se.
>
> - **Clarified contribution**: Emphasized the theoretical grounding and systematic generation mechanism as our core innovations.
>
> ### Major Weakness 2: Transitive Closure vs. Direct Causal Relationships
>
> **The concern**: Recovering $G^+$ (transitive closure) includes indirect relationships, while the goal should be the true DAG $G$ with direct relationships only.
>
> **Our response**: This concern is addressed in our overall response (Section 1), but we emphasize:
>
> **Our position**: The transitive closure is a **stronger intermediate target** than what single-ordering methods achieve. All ordering-based methods require pruning - but we provide a much better foundation for that pruning step.
>
> **Manuscript changes**:
>
> - **Enhanced discussion**: Made the role of CAM pruning more central to the methodology description.
>
> - **Explicit positioning**: Stated that the ultimate goal is $G$, but that $G^+$ provides superior intermediate representation.
>
> - **Limitation acknowledgment**: Clarified that DOTS inherits CAM pruning limitations but provides better input.
>
> ### Major Weakness 3: Diversity of Orderings from Different Noise Scales
>
> **The concern**: The connection between noise scales k and truly diverse/complementary orderings needs more rigorous substantiation beyond high-level intuition.
>
> **Our response**: **This has been addressed with both theoretical and empirical analysis**.
>
> **Manuscript changes implemented**:
>
> - **Corrected frequency domain analysis**: Now provides rigorous mathematical foundation showing how different k values create variable-bandwidth filtering, yielding diverse frequency emphasis.
>
> - **Kendall tau analysis**: Added empirical validation measuring ordering diversity across different k values, demonstrating they are complementary rather than minor variations.
>
> - **Extended ablation**: Enhanced Figure 9 analysis to show performance scaling with ordering diversity, not just quantity (Figure 16).
>
> **Theoretical justification**: The corrected mathematical framework establishes formal conditions under which multi-scale sampling provides complementary structural information.
>
> ### Major Weakness 4: Unfair Baseline Comparisons
>
> **The concern**: Comparisons with adapted static methods ("-C" suffix) are unfair since these weren't designed for temporal data. Real-world results show closer performance with temporal methods.
>
> **Our response**: **This is a valid and important critique**.
>
> **Manuscript changes**:
>
> - **Enhanced experimental discussion**: Added explicit caveats that adapted static methods are not designed for temporal settings.
>
> - **Enhanced exposition**: We make it clearer which methods belong to which category (temporal or non-temporal) and emphasise that comparing to temporal methods is our priority.
>
> - **Focused comparisons**: Emphasized comparisons with temporal methods (PCMCI+, TiMINo, VARLiNGAM) as our primary evaluation.
>
> **Fair assessment**: The reviewer is correct that strong performance, relative to adapted methods, is less meaningful. Our value is demonstrated through competitive performance with methods designed for temporal data.

---

> > ### Author Response · Authors · 2025-09-11
> > **Response to Reviewer t6jd (2/2)**
> >
> > ### Minor Issues Addressed
> >
> > **Diffusion model justification**
> >
> > **Manuscript changes**: Added comprehensive discussion in Section 4 comparing diffusion models to alternatives (normalizing flows, etc.), highlighting specific advantages:
> >
> > - **Scalability**: Better scaling to high-dimensional lag-embedded data.
> >
> > - **Stability**: More stable training compared to adversarial approaches.
> >
> > - **Multi-scale property**: Natural mechanism for diverse ordering generation.
> >
> > **Enhanced ablation study**
> >
> > **Manuscript changes**: Extended Figure 9 to include comparison between aggregated results and best individual ordering performance, demonstrating benefits beyond simply getting more "shots" at success (see Figure 16).
> >
> > ### Summary of Changes
> >
> > **Honest positioning**: Reframed as theoretically motivated ensemble approach rather than claiming fundamental aggregation novelty.
> >
> > **Experimental transparency**: Added appropriate caveats about baseline fairness and acknowledged superior baseline performance on specific datasets.
> >
> > **Theoretical strengthening**: Provided rigorous mathematical foundation for multi-scale diversity claims.
> >
> > **Technical justification**: Enhanced diffusion model motivation with detailed comparison to alternatives.
> >
> > The revised manuscript maintains our core contributions while addressing the reviewer concerns regarding positioning, experimental fairness, and theoretical rigor.

---

### Review · Reviewer_Tqog · 2025-08-27

**Summary Of Contributions:**

This paper proposes a diffusion model-based multi-level linear extension based method to estimate the topological closure of the temporal DAG of a multi-time lag dynamic Bayesian network with contemporaneous relations. It provides empirical evaluations demonstrating improved performance above baselines in both synthetic and real world data sets.

**Audience:**

Yes

**Broader Impact Concerns:**

No ethical/impact concerns

**Claims And Evidence:**

No

**Requested Changes:**

- **Critical:** The theoretical exposition needs to be **corrected**.
- **Critical:** Please explain, in your response, why the authors chose to jump directly to time-series causal discovery and not also claim that their approach is useful for plain causal discovery as well. Depending on their own framing, I expect the relevant claims and/or clarifications to be incorporated in the revision. A logical leap to this extent weakens my trust in the correctness of other parts of this paper.
- **Critical:** Adding on top of this, _if_ the method is applicable to causal discovery as well, then comparisons on this setting as well. Specifically, add synthetic and real-data experiments comparing against non-time-series methods.
- I don't expect a full-scale theoretical analysis of the proposed approach to be provided in the next revision, but I would like it if the authors manage to do so.
- Restructure the assumptions. See weaknesses for details.
- Add the Gaussianity assumption on exogenous noises to the set of assumptions. You mostly use results/methods of (Sanchez et al., 2023) which requires this AFAIK.
- Other two formatting issues are mostly suggestions. I expect the authors to accommodate them as much as possible, but they are not critical for my acceptance by themselves.
- Add a reference to Hiraguchi's theorem: For all partial orders, there exists $\lceil n/2 \rceil$ total orders that fully specify it, i.e., the "valid edge set" equals the transitive closure using graph terminology. This observation -- while not directly relevant when sampling the total orders randomly -- justifies the multi-total order framework very strongly.
- Verify that recovering the topological order too is NP-hard. Chickering (1994) probably already implies this, but add another citation that explicitly states it.
- Section 3.2, using limit in that equation is slightly overcomplicating it. It looks fine, but there is probably a more professional and cleaner way to describe it.
- Figure 3 lacks the experiment setting description, and its explanation is spread across two subsections. Tighten this and add missing context.
- Minor typo on pg. 11: it should be $\mathbb{D} \in \mathbb{R}^{(T - \tau\_{max}) \times d \times (\tau\_{max} + 1)}$ ($\tau$'s missing the $+1$ currently)
- In many places, $k$ and $\alpha\_k$ are conceptually confused with each other. Proofread and make them consistent.
- Rolland (2022) reference is duplicated
- Pg. 13: Description of how you reach $T =19 200$ samples overall is slightly vague. Can you clarify?
- Pg. 14: CAM is not a score-matching method; it predates (Rolland 2022).
- You have included CAM already, you can add additional plain causal discovery algorithms here as well for comparison. Adding at least 1 or 2 more is important for completeness, adding more state-of-the-art models would help your case.
- The summary graph formulation doesn't fit well with the proposed algorithm's use cases. Please justify how they fit or remove/move to appendix otherwise.

**Strengths And Weaknesses:**

**Strengths.** The main strength of the paper is its main idea: Multiple total orders are more useful than a single one, while also being cheaper than obtaining the partial order. Very neat observation that could be useful across many causal discovery tasks -- or any DAG learning task in general too. The approach is well-motivated, the way of sampling multiple total orders is very intriguing, and algorithm is overall well-presented and clean.

**Weaknesses.**
There are multiple _classes_ of weaknesses. First, **theory problems**:
- The masking-based causal discovery method (Sanchez et al., 2023) operates on the true data distribution. Therefore, in the noisy data model the authors use, the trick used to derive the score of the marginal distributions using Hessians **is not** (necessarily) **valid**. I acknowledge that the paper is mostly an algorithmic contribution, regardless, this constitutes a big gap in the "correctness" of the proposed algorithm.
- Frequency domain explanation (Section 4.1): Fourier transformation is taken incorrectly. Density of sum of independent variables is the convolution of individual densities, Fourier transform of which is the product. This tracks, since adding independent Gaussian noise is equivalent to applying a Gaussian filter to the densities. While this is still a variable-width filter applied on the "frequency" data and therefore the high level description of the procedure still is well-founded, the technical details are simply **incorrect**. Similarly for Fig. 5.
- Overall, the proposed framework is "motivated" and not really "proved" with any rigor. Given how related literature generally provides correctness statements together with the algorithms, it would help the paper to have them as well to situate it more strongly in the literature.
- The paper, as it stands, does not make it clear whether said proofs exist or not. For example, having formal assumption statements like in Section 2.2 usually suggests -- but does not *directly* imply -- there is a formal theorem using them somewhere. Having a "theory" section implies similarly. These are not wrong per se, but the language needs to be adjusted so that readers don't get the wrong idea.

Second class is **presentation/framing** issues:
- While the idea of using multiple total orders at the same time is a very neat concept, I do not see the reason why the authors made the jump of going directly to time series modeling rather than plain causal discovery task. Yes, the idea of using the temporal dependence structure to further refine the edge set is a neat trick, but it is literally the only additional step beyond the causal discovery algorithms. More constructively, why not present the causal discovery variant as well?
- The paper belabors graph theory/terminology for a long time. These are well established concepts, simply use them. You can add references to relevant _textbooks_ if need be, but please don't overexplain.
- The literature on score estimation could be consolidated. I liked how the authors tailor each nuanced discussion to the context, but this also causes some repetition and back-and-forth when trying to see what is available at all. You could try lightening in-text descriptions in favor of extending the relevant related work section.

On experiments.
- The summary graph formulation doesn't fit well with the proposed algorithm's use cases. Please justify how they fit or remove/move to appendix otherwise.
- In page 14, CAM is not a score-matching method; it predates (Rolland 2022).
- You can add additional plain causal discovery algorithms for comparison. Adding at least 1 or 2 more is important for completeness, adding more state-of-the-art models would help your case.

---

> ### Author Response · Authors · 2025-09-11
> **Response to Reviewer Tqog**
>
> **Strengths acknowledged**: Thank you for recognizing the value of our multi-ordering concept and its potential broader applicability, as well as acknowledging our approach as "well-motivated" with "very neat observation."
>
> We address your concerns systematically:
>
> ### Critical Issue 1: Theoretical Exposition Corrections
>
> **Masking-based approach validity in noisy data models**:
>
> A potential gap, regarding the validity of Hessian-based score derivation in noisy settings, is correctly identified. We clarify:
>
> - **Sanchez et al. (2023) framework**: The original DiffAN operates under Additive Noise Models (ANMs): $x_j = f_j(pa_j) + ε_j$, which are inherently noisy data models. The masking-based approach was specifically designed and validated for this noisy setting.
>
> - **Our temporal extension**: We extend to temporal ANMs (TiMINo framework) where $x_j^t = f_j(pa_j^{t}) + ε_j^t$. The theoretical validity is preserved as each time-indexed variable follows the same ANM structure.
>
> - **Score derivation validity**: The Hessian-based approach relies on noise regularity conditions ($\partial^2 \log p^u / \partial x^2$ = constant) that hold for Gaussian noise and other distributions with quadratic log-densities. Similar to Sanchez et al. (2023), our approach operates under the same theoretical framework. We add this assumption in Assumption 5 in the revised manuscript.
>
> **Frequency domain analysis correction**:
>
> The reviewer is absolutely correct about the mathematical error. We have corrected the derivation to operate on densities/characteristic functions rather than signals. For $x_k = \sqrt{\alpha_{k}}  x_0 + \sqrt{1-\alpha_{k}} \epsilon$ with $x_0$ ⊥ $\epsilon$:
>
> - Density is convolution, characteristic function factorizes: $\phi_{x_k}(\omega) = \phi_{\sqrt{\alpha_k} x_0}(\omega) \cdot \phi_{\sqrt{1-\alpha_k} \epsilon}(\omega) = \phi_{x_0}(\sqrt{\alpha_k} \omega) \cdot \phi_{\epsilon}(\sqrt{1-\alpha_k} \omega)$
>
> - For Gaussian $\epsilon$: becomes Gaussian low-pass multiplier $\exp(-\frac{1-\alpha_k}{2}|\omega|^2)$
>
> - This yields variable-bandwidth smoothing as k increases
>
> **Theoretical rigor and formal statements**:
>
> The reviewer raises an important point about the mismatch between formal assumption statements and the lack of accompanying theorems. We have addressed this by softening the theoretical language and claims:
>
> - Changing "Theory" to "Theoretical Motivation" (Section 3)
>
> - Modifying "Proof" to "Justification"
>
> - Clearly distinguishing what we prove rigorously versus what we build upon from established theory
>
> ### Critical Issue 2: Why Temporal Focus Rather Than General Causal Discovery?
>
> This is a fundamental question about our scope which we address in the overall response to common concerns.
>
> ### Critical Issue 3: Experimental Additions
>
> **Regarding static causal discovery experiments**: Given our honest assessment that temporal advantages are fundamental to our approach, adding static experiments would not strengthen our case and might mislead readers about our method scope.
>
> **Summary graph formulation**: Summary graph is the output of the TiMINo framework, which is an important temporal causal discovery method. We use the summary to enable this benchmark comparison.
>
> ### Additional Changes Implemented
>
> **Theoretical references**:
>
> - **Hiraguchi's theorem**: Added explicit citation showing that aggregating linear extensions of a partial order recovers the full partial order, providing formal justification for our multi-ordering framework.
>
> - **NP-hardness**: Added citation confirming topological ordering recovery complexity.
>
> **Assumptions restructuring**:
>
> - Added formal Gaussianity assumption on exogenous noise (Assumption 5).
>
> - Connected assumptions explicitly to Sanchez et al. (2023) framework.
>
> - Clarified noise distribution regularity conditions.
>
> **Presentation improvements**:
>
> - Streamlined graph theory terminology as suggested.
> - Improved Figure 3 experimental context.
> - Fixed notation consistency issues and typos.
> - Clarified sample generation description.
> - Corrected statement about CAM predating score-matching methods. We meant that CAM is an ordering-based method.
>
> The revised manuscript now provides clearer theoretical exposition, honest scope assessment, and improved presentation while addressing the specific concerns about mathematical rigor and methodological positioning.

---

### Review · Reviewer_eFk5 · 2025-08-28

**Summary Of Contributions:**

This paper introduces DOTS, a novel method for temporal causal discovery. The core contribution is a paradigm shift from traditional methods that rely on a single, often fragile, causal ordering. Instead, DOTS proposes to generate and aggregate *multiple* valid causal orderings to enhance the robustness and accuracy of structure learning. Theoretically, the authors argue that by integrating information from multiple orderings, their method effectively recovers the **transitive closure** ($G^+$) of the underlying Directed Acyclic Graph (DAG), which helps mitigate spurious artifacts inherent in single-ordering approaches. To achieve this, the paper presents a key technical innovation: the use of a **multi-scale denoising diffusion model**. This allows for the efficient sampling of a diverse set of causal orderings from a single trained model by leveraging different noise scales, which are argued to correspond to different frequency components of the data. Extensive experiments on different benchmark datasets demonstrate that DOTS outperforms state-of-the-art baselines in terms of causal structure accuracy and offers a scalable solution to the problem.

**Audience:**

Yes

**Broader Impact Concerns:**

The research presented in this paper raises no ethical concerns or broader impact issues.

**Claims And Evidence:**

Yes

**Requested Changes:**

1. The paper makes strong claims about the robustness of DOTS. Recent large-scale benchmark studies (Montagna et al., 2023), have demonstrated that violations of these assumptions are critical failure points for many causal discovery algorithms, often leading to performance comparable to a random baseline. However, the experiments do not test the method's performance against critical and common violations of its underlying assumptions, such as non-stationary processes. It is critical that the authors add a dedicated discussion in the limitations or experiments section to:
    * Acknowledge that the current empirical evaluation does not cover scenarios where these assumptions are violated.
    * Contextualize their robustness claims accordingly, clarifying that they pertain to variations in sample size and dimensionality rather than fundamental assumption violations.

2. The core innovation (diffusion-based multi-ordering aggregation) recovers the transitive closure ($G^+$), which is fundamentally unable to distinguish direct from indirect causes. The final, crucial step of producing a sparse DAG is entirely delegated to an external, heuristic method (CAM pruning). The authors should clarify this in the methodology and discussion sections, explicitly stating that the final performance is a hybrid of their novel framework and the chosen pruning module, and that DOTS inherits the limitations of that module.

3. The soft-voting threshold, $\theta$, is a critical hyperparameter that directly controls the precision-recall trade-off of the resulting graph. The paper would be significantly strengthened by an ablation study that shows how the F1 score, precision, and recall metrics vary with different values of $\theta \in (0, 1]$. This would provide invaluable practical guidance for users and a deeper understanding of the method's behavior.

4. The related work section provides a good overview, but it would be strengthened by positioning the paper's contributions in the context of several highly relevant, recent advancements in score-matching/diffusion-based causal discovery. We suggest the authors add and discuss the following works:
- The work of Chen et al. (2024) is directly and critically relevant, as it also proposes a scalable, score-matching-based method specifically designed for temporal data. This paper presents an alternative strategy for improving efficiency.
- The paper should discuss recent efforts to generalize the score-matching framework. For instance, the work of Montagna et al. (2023a) introduces an algorithm for models with arbitrary noise distributions, while Liu et al. (2024) addresses scalable discovery in mixed linear and nonlinear models.
- Any claims of robustness should be contextualized by the findings in Montagna et al. (2023b). This paper provides a large-scale benchmark on how score-matching methods perform when core assumptions (like causal sufficiency) are violated.


**References**

Chen, H., Yi, K., Liu, L., & Wang, Y. G. (2024). Score-matching-based Structure Learning for Temporal Data on Networks. arXiv preprint arXiv:2412.07469.

Liu, W., Huang, B., Gao, E., Ke, Q., Bondell, H., & Gong, M. (2024). Causal Discovery with Mixed Linear and Nonlinear Additive Noise Models: A Scalable Approach. Proceedings of the Third Conference on Causal Learning and Reasoning (CLeaR), PMLR 236:1237-1263.

Montagna, F., Noceti, N., Rosasco, L., Zhang, K., & Locatello, F. (2023a). Causal Discovery with Score Matching on Additive Models with Arbitrary Noise. Proceedings of the 2nd Conference on Causal Learning and Reasoning (CLeaR), PMLR 213:726-751.

Montagna, F., Mastakouri, A. A., Eulig, E., Noceti, N., Rosasco,L., Janzing, D., Aragam, B., & Locatello, F. (2023b). Assumption violations in causal discovery and the robustness of score matching. Advances in Neural Information Processing Systems (NeurIPS), 36.

**Strengths And Weaknesses:**

**Strengths**

- The paper's core idea to aggregate multiple causal orderings, rather than relying on a single one, is a significant and innovative contribution. This approach directly addresses the fragility of traditional single-ordering methods, which can introduce spurious artifacts.
- The method cleverly employs a multi-scale diffusion model to efficiently generate a diverse set of causal orderings from a single trained network. The justification that different noise scales emphasize different frequency components of the data provides a compelling rationale for the diversity of the generated orderings.
- The paper is well-written, clearly structured, and easy to follow. The methodology is explained logically, and figures such as the pipeline diagram (Figure 4) effectively illustrate the algorithm's workflow.

**Weaknesses**
- The theoretical framework aims to recover the transitive closure ($G^+$) of the DAG, which by definition cannot distinguish between direct and indirect causal relationships. This makes the method heavily dependent on a final, heuristic pruning step (CAM pruning) to obtain the final sparse graph, and the overall performance is thus a hybrid of the novel framework and this external module.
- Training diffusion models, especially on the high-dimensional lag-embedded data required by DOTS, can be computationally very expensive. Furthermore, the method's performance is sensitive to key hyperparameters, such as the voting threshold $\theta$ and the maximum lag $\tau_{max}$, for which the paper provides limited practical guidance on selection.
- The method relies on strong assumptions such as causal sufficiency and stationarity. While this is standard for the field, the paper does not empirically evaluate the method's performance in scenarios where these critical assumptions are violated (e.g., in the presence of hidden confounders or non-stationary data), making its real-world robustness an open question.

---

> ### Author Response · Authors · 2025-09-11
> **Response to Reviewer eFk5 (1/2)**
>
> **Strengths acknowledged**: We appreciate your recognition of our core innovation ("significant and innovative contribution"), the multi-scale diffusion approach ("cleverly employs"), and the clear presentation of our methodology ("well-written, clearly structured").
>
> We address your specific concerns systematically:
>
> ### Requested Change 1: Robustness Claims and Assumption Violations
>
> **The concern**: It is correctly pointed out that we make strong robustness claims without empirically evaluating performance under assumption violations (non-stationarity, hidden confounders), referencing Montagna et al. (2023b) findings that such violations can lead to random baseline performance.
>
> **Our response**: We acknowledge that our robustness claims need proper contextualization.
>
> **Manuscript changes implemented**:
>
> - **Added comprehensive limitations section (Section 5.7)** explicitly acknowledging:
>
>   - Current evaluation scope is limited to stationary, causally sufficient settings.
>
>   - Our robustness claims pertain specifically to variations in sample size/dimensionality, not fundamental assumption violations.
>
>   - Need for future work on non-stationary and confounded settings.
>
> - **Contextualized robustness claims throughout** to clarify they refer to:
>
>   - Robustness to small sample sizes and high dimensionality.
>
>   - Stability across different lag structures and nonlinearity levels.
>
>   - **NOT** robustness to fundamental assumption violations.
>
> ### Requested Change 2: Transitive Closure and Pruning Step Dependency
>
> **The concern**: The method recovers $G^+$ (transitive closure) which cannot distinguish direct from indirect relationships, making final performance a "hybrid" dependent on external CAM pruning.
>
> **Our response**: This is addressed comprehensively in our overall response (Section 1), but we emphasize:
>
> **Key clarification**: This limitation is **not unique to our method**. All ordering-based methods (CAM, SCORE, DAS, NoGAM, DiffAN) require pruning steps to obtain sparse DAGs. However, our contribution provides a **much stronger foundation** for pruning:
>
> - Single orderings recover weak approximations with many spurious edges.
>
> - Our transitive closure recovery provides the theoretically correct intermediate representation.
>
> - The hybrid nature is thus an **advantage** over single-ordering approaches.
>
> **Manuscript changes**:
>
> - **Enhanced methodology section** to explicitly state the hybrid nature and positioning CAM as integral rather than external.
>
> - **Added discussion** clarifying that DOTS inherits CAM limitations but provides superior input for the pruning step.
>
> - **Emphasized** that the transitive closure is a stronger theoretical guarantee than previous single-ordering methods achieve.
>
> ### Requested Change 3: Soft-Voting Threshold Ablation Study
>
> **The concern**: The soft-voting threshold $\theta$ critically controls precision-recall trade-offs but lacks comprehensive ablation analysis.
>
> **Our response**: **This has been implemented** - we have added comprehensive ablation studies.
>
> **Manuscript changes**:
>
> - **Section C.2**: Added detailed ablation study showing F1/precision/recall metrics across different $\theta$ values.
>
> - **Practical guidance**: Provided recommendations for $\theta$ selection based on desired precision-recall trade-offs.
>
> - **Analysis**: Demonstrated how $\theta$ affects spurious edge filtering and true edge retention.
>
> This provides the "invaluable practical guidance", as requested, for users.
>
> ### Requested Change 4: Enhanced Related Work
>
> **The concern**: Need to position our work within recent score-matching/diffusion-based causal discovery advances, specifically Chen et al. (2024), Montagna et al. (2023a,b), and Liu et al. (2024).
>
> **Our response**: **Comprehensive additions implemented**.
>
> **Manuscript changes**:
>
> - **Chen et al. (2024)**: Added discussion of their alternative score-matching strategy for temporal data, highlighting complementary approaches to efficiency.
>
> - **Montagna et al. (2023a)**: Discussed their generalization to arbitrary noise distributions within the score-matching framework.
>
> - **Liu et al. (2024)**: Covered their scalable approach to mixed linear/nonlinear models.
>
> - **Montagna et al. (2023b)**: **Critically important** - used their robustness findings to properly contextualize our robustness claims, acknowledging assumption violation challenges.

---

> ### Author Response · Authors · 2025-09-11
> **Response to Reviewer eFk5 (2/2)**
>
> ### Summary of Changes
>
> **Theoretical positioning**: Maintained honest assessment of contributions while better contextualizing within recent literature.
>
> **Experimental transparency**: Added comprehensive ablations and acknowledged limitation scope explicitly.
>
> **Claims moderation**: Robustness claims now properly contextualized to sample size/dimensionality rather than fundamental assumption violations.
>
> **Literature integration**: Comprehensive integration of suggested references with detailed discussion of relationships to our work.
>
> The revised manuscript addresses all your specific concerns while maintaining scientific rigor and honest assessment of our method's capabilities and limitations.

---

### Author Response · Authors · 2025-09-11
**General response to all reviewers (1/2)**

We thank all three reviewers for their thorough and constructive feedback. We appreciate the recognition of our core contribution—using multiple causal orderings to enhance robustness in temporal causal discovery—as well as the detailed theoretical critiques that will help to strengthen our work. Below, we provide both a comprehensive response to common concerns and individual responses to each reviewer. All acknowledged changes are highlighted in the updated manuscript and will be finalised for the camera-ready as appropriate.

Several key concerns were raised across multiple reviews. We address these systematically:

### 1. Why Focus on the Transitive Closure $G^+$ for theoretical motivation?

**The concern**: Multiple reviewers (eFk5, t6jd) questioned why we target the transitive closure $G^+$ instead of the direct causal graph $G$, noting that $G^+$ includes indirect relationships.

**Our response**: This design choice is both theoretically motivated and practically advantageous:

- **Stronger theoretical guarantee**: Recovering the transitive closure $G^+$ is a much stronger claim than what has been previously achieved by single ordering methods (CAM, SCORE, DAS, NoGAM, DiffAN). A single ordering recovers only a weak approximation of the causal structure because, for any given position in the ordering, all subsequent positions are considered as potential descendants, introducing many spurious edges.

- **Principled pruning workflow**: All ordering-based methods require a pruning step to obtain the final sparse DAG—this is not unique to our approach. However, by first recovering the theoretically grounded transitive closure, we provide a stronger foundation for subsequent pruning than single-ordering methods can achieve.

- **Robustness through aggregation**: The transitive closure provides a more stable target for aggregation across multiple orderings, as it captures the complete reachability structure of the DAG. This stability is crucial for our multi-ordering approach.

- **Temporal context advantages**: In temporal data, the distinction between direct and indirect effects becomes more nuanced due to lag structures, where an indirect effect at one lag may represent a direct effect at a longer lag. The transitive closure naturally captures this temporal complexity. (see next point)

### 2. Why Focus on Temporal Causal Discovery?

**The concern**: Reviewers questioned why we focus specifically on temporal causal discovery rather than general causal discovery, asking why we do not also claim broader applicability.

**Our response**: The temporal setting provides unique theoretical and practical advantages that make our approach particularly well-suited:

**Theoretical advantages**:

- **Enhanced identifiability**: Temporal data provides natural constraints through the temporal priority principle (causes precede effects), which significantly reduces the search space and enhances identifiability compared to static causal discovery.

- **Natural multi-scale structure**: Time series data inherently contains multi-scale temporal dependencies that naturally align with our diffusion-based multi-scale approach.

- **Reduced spurious edges**: Temporal precedence provides a natural filtering mechanism that eliminates backward-in-time connections.

**Practical advantages**:

- **Computational efficiency**: Our approach leverages temporal structure for efficiency gains that are not available in static settings.

- **Natural ordering constraints**: Without temporal precedence, generating diverse orderings would require different mechanisms that may not be as computationally efficient.

**Honest scope assessment**: While the core insight about aggregating multiple orderings could potentially apply to static causal discovery, the specific advantages of our approach are fundamentally tied to the temporal setting. Making broader claims without proper validation would be misleading.

---

> ### Author Response · Authors · 2025-09-11
> **General response to all reviewers (2/2)**
>
> ### 3. Enhanced Empirical Validation Following Review Feedback
>
> **The concern**: Reviewers requested additional ablation studies and empirical validation to support our claims.
>
> **Our response**: We have conducted comprehensive additional experiments addressing all reviewer requests:
>
> **New ablation studies implemented**:
>
> - **Soft-voting threshold analysis**: Added comprehensive ablation study showing F1/precision/recall trade-offs across different $\theta$ values (Section C.2), providing practical hyperparameter guidance.
>
> - **Maximum lag analysis**: Added analysis of maximum lag parameter effects and computational trade-offs (Section C.1).
>
> - **Ordering diversity validation**: Used Kendall tau correlation analysis to quantify the diversity of orderings generated at different diffusion noise scales k, demonstrating that different scales produce truly diverse and complementary orderings rather than minor variations (Section C.3).
>
> - **Multi-ordering contribution**: Extended existing Figure 9 analysis to show how performance changes with the noise scale value at individual levels, confirming the theoretical benefits of aggregation (Figure 16).
>
> These additions provide the empirical foundation requested by reviewers as well as add some additional valuable insight on our method's behaviour, while maintaining scientific honesty about the method's scope and limitations.
>
> ### 4. Transitive Closure vs. Direct Causal Relationships
>
> **Concern**: Multiple reviewers (eFk5, t6jd) highlighted that recovering the transitive closure $G^+$ includes indirect relationships, which may not align with the goal of finding direct causal relationships.
>
> **Response**: We acknowledge this important distinction and will clarify our theoretical framework in the revision. Our approach deliberately targets the transitive closure as an intermediate representation for the following reasons:
>
> - **Robustness**: The transitive closure provides a more stable target for aggregation across multiple orderings, as it captures the full reachability structure of the DAG.
>
> - **Temporal context**: In temporal data, distinguishing between direct and indirect effects becomes more nuanced due to lag structures, where an indirect effect at one lag may represent a direct effect at a longer lag.
>
> - **Practical workflow**: The final sparse DAG is obtained through a principled pruning step (CAM), which we position as an integral part of our methodology rather than an external heuristic.
>
> **Manuscript changes**: We have:
>
> - Added explicit discussion in Section 3.1 acknowledging that $G^+$ contains indirect relationships (new clarification box after Proposition 1).
>
> - Clarified that our method provides a robust intermediate representation that is then refined to recover direct relationships.
>
> - Emphasized the hybrid nature of our approach and positioned the pruning step as theoretically motivated within our framework (revised CAM pruning section).

---

### Decision · Action_Editor_W113 · 2025-10-30

**Recommendation:** Accept as is

**Additional Comments:**

I have not requested any revision. But the changes in blue and the version that was uploaded latest must be adopted with additional checks for grammar and typos.

**Audience:**

Yes

**Audience Explanation:**

The paper applied diffusion style denoising to estimate hessian of log likelihoods and uses the information at different noise scales to enable causal discovery for time series. This is a valuable contribution at the intersection of causality and diffusion.

**Claims And Evidence:**

Yes

**Claims Explanation:**

Summary:
  Paper used denoising score matching to estimate scores for time series data and then uses its Jacobian to identify leaf nodes in the unknown causal ordering that underlies time series data. The notion of using Jacobian of scores to identify leaf nodes in additive noise models is well known since Rolland et. al. 2022.

The paper's main novelty is to apply this to time series data with suitable modifications followed by an interesting empirical insight that estimating Jacobians at different noise scales gives different ordering estimates which are aggregated into one transitive closure that can capture interaction at different frequency scales. Experiments show its value on time series causal discovery task with nice comprehensive comparisons to a lot of good baselines.

Main concerns:
 1) There were some corrections pointed out on some formal claims
 2) Why cant this be applied to static causal discovery problems ? (not the temporal case)



Authors corrected these claims to the satisfaction of the reviewers. Point number 2 - authors clarified that they would stick to their current scope of temporal causal discovery.

 Almost every reviewer is in agreement that while novelty with respect to the central idea used in not very big, the paper neverthless contributes to time series causal discovery by applying diffusion style denoising to this problem and a novel aggregation technique at different noise scales to capture interactions across time robustly.

I find the experimental results compelling - particularly with strong methods like PCMCI etc as another reviewer pointed out. This shows that Jacobian of scores can enable robust causal discovery for time series data and its a valuable contribution establishing the efficacy of such ideas for temporal causal discovery.

Recommendation:Accept